



# Relative contribution of stand characteristics on carbon stocks in subtropical secondary forests in Eastern China

Arshad Ali[1,2], En-Rong Yan[1,2,*], Han Y. H. Chen[3], Yan-Tao Zhao[1,2], Xiao-Dong Yang[1,2], and Ming-Shan Xu[1,2]

[1] School of Ecological and Environmental Sciences, East China Normal University, Shanghai 200241, China;

[2] Tiantong National Forest Ecosystem Observation and Research Station, Ningbo 315114, Zhejiang, China;

[3] Faculty of Natural Resources Management, Lakehead University, 955 Oliver Road, Thunder Bay, ON P7B 5E1, Canada

[*] Author for correspondence

Tel & Fax: +86-21-54341164

Email: *eryan@des.ecnu.edu.cn*

**Running title:** Stand characteristics and forest carbon stocks

Text pages: (without references, and Tables and Figures): 18

Tables: 2          Figures: 3          References: 58

Supplementary information: Tables: 14          Figures: 5

**Contribution of the co-authors:** AA and ERY conceived and designed the study. ERY coordinated the research project. AA, YTZ, XDY and MSX conducted sampling design, field and lab works. AA analyzed the data. AA and ERY wrote the paper. HYHC reviewed, commented and edited the paper draft. All the authors read and approved the final manuscript.



## Abstract

Stand structural diversity, which is characterized by species diversity, variances in tree diameter at breast height (DBH) and height, plays an important role in influencing forest carbon (C) stocks. However, the relative contribution of stand structural diversity in contrast to other stand characteristics on the variation in C stocks in subtropical forests have not been fully explored. In this study, aboveground C stock, soil organic C stock, tree species, DBH and height diversities, stand age, and stand density, and site productivity were determined across 80 subtropical forest plots in Eastern China. Using simple regression analysis, we found that DBH and height diversities, site productivity, and stand age explained 49%, 13%, 41%, and 50% of the variation in aboveground C stock, respectively, whereas species diversity and stand density did not explained any variation (i.e., < 1%). Multiple regression analysis indicated that variation in aboveground C stock was explained to a higher degree (83%) by the joint effects of DBH diversity, stand age, site productivity, species diversity and height diversity than by stand structural diversity (54%), and the other three stand characteristics (79%) alone. The structural equation modelling (SEM) showed that the effect of stand age on aboveground C stock was stronger directly (beta = 0.59) than indirectly (beta = 0.11). Stand age has also significant and strong effect on DBH (beta = 0.63) and height (beta = 0.55) diversities. Six stand characteristics did not explain any variation in soil organic C stock (i.e., < 2%), based on both simple and multiple regressions analyses, as well as SEM analysis. Our analyses suggest that, rather than species and height diversities, DBH diversity, stand age and site productivity cumulatively contributed to variation in aboveground C stock during stand development in subtropical secondary forests in Eastern China. Therefore, improving tree DBH diversity and stand condition could be an effective approach for enhancing C storage in subtropical forests.



**Key words:** biodiversity; carbon storage; evergreen broadleaved forests; forest structure;

regressions; structural equation model.





# 1    Introduction

Subtropical forests in the East Asian monsoon region play a critical role in global carbon (C)

cycling, and store more C than previously thought (Yu et al., 2014). Currently, most of these

forests are naturally regenerated secondary forests (Wang et al., 2007), and their C stocks

increase as they recover from disturbances (Yu et al., 2014). Despite their importance, we

still lack a complete understanding of how C stocks vary with changes in stand characteristics

in these forests. It is well known that biomass or C stocks in forest ecosystems are directly

impacted by site productivity (Lohbeck et al., 2015), stand density (Vayreda et al., 2012), tree

species diversity (Con et al., 2013), tree diameter at breast height (DBH, diameter at 1.5 m

above root collar) diversity, and tree height diversity (Wang et al., 2011) (Fig. 1). The last-

three diversity parameters alone or combined are typically defined as the stand structural

diversity (Staudhammer and LeMay, 2001). In addition, stand age, as an indicator for stand

development following disturbances, has been identified as a primary factor that influences

aboveground biomass (AGB) in both even-aged (Böttcher et al., 2008) and naturally uneven-

aged (Becknell and Powers, 2014) forest stands.  Moreover, variabilities in stand structural

diversity, site productivity, and stand density depend to a large degree on stand age (Lei et al.,

2009; Wang et al., 2011; Lohbeck et al., 2015). Therefore, stand age may affect C stocks

indirectly through the alteration of other stand characteristics, such as stand structural

diversity, site productivity, and stand density in forest ecosystems (Fig. 1).

There has been a reinvigorated research interest in analyzing how AGB (thus

aboveground C stock) vary with stand age, species composition, and abiotic factors, in both

managed plantations (Smith et al., 1997) and natural secondary forests (Clark and Clark,

2000; Becknell and Powers, 2014); however, discrepancies among studies remain unresolved.

For instance, some studies have documented that the relationship between species diversity





and AGB was either positive (Wang et al., 2011; Con et al., 2013; Zhang and Chen, 2015; Dayamba et al., 2016), negative (Szwagrzyk and Gazda, 2007), or non-significant (Vilà et al., 2003). The relationship between species diversity or richness and soil resident organic C has also been reported to be either positive, in an old-growth forest in Northeast China (Chen,

2006), in a boreal forest in northern Sweden (Jonsson and Wardle, 2009), and under different land use types in tropical West Africa (Dayamba et al., 2016), or non-significant in a subalpine coniferous forest (Zhang et al., 2011).

In addition to species diversity, forest productivity and aboveground C stock are also related to many other factors such as tree size inequality, stand age, nutrients regime, and

climate anomalies (e.g., Chen and Luo, 2015; Zhang and Chen, 2015). Empirical studies have demonstrated that aboveground C was either related to stand structural diversity, site productivity, or stand age in tropical forests (e.g., Wang et al., 2011; Con et al., 2013; Becknell and Powers, 2014; Stephenson et al., 2014; Lohbeck et al., 2015; Poorter et al., 2015). Changes in stand characteristics through forest succession have significant impacts on

forest productivity and aboveground C stock (Becknell and Powers, 2014). This is because tree size inequality among and within species are critical toward maintaining species, DBH and height diversities (collectively referred as "stand structural diversity"; Wang et al., 2011), which has been recognized to significantly affect forest C stocks (Lexerød and Eid, 2006; Zhang and Chen, 2015). It is understandable that stand structural diversity is shaped by

species composition with different sized (DBH and height) trees in multistory canopies (Liang et al., 2007; Lei et al., 2009). At the community level, variations among tree diameters and heights, resulting from both differences within and among species (Zhang and Chen, 2015), may allow different levels of tree canopy heights, and increase the C synthesis of sub-canopy trees or understory plants by facilitating an increase in the availability of light (Chave

et al., 2005).



Even though the bulk of evidence suggests that forest C stocks are ecologically linked to stand structural diversity, stand productivity, stand density and age in other forest ecosystems, it remains unclear how stand structural diversity alone, or in combination with stand age, site productivity and density, explain the variation in C stocks in secondary

subtropical forests. Recently, Barrufol et al. (2013) found that Chinese subtropical tree diversity is an important driver of forest productivity and re-growth after disturbance that supports the provision of ecological services. However, field tests of which stand characteristic best explain variations in C stocks are rarely done (but see Wang et al., 2011; Con et al., 2013), and remains unclear in secondary subtropical forests. In this context, we

anticipated that stand structural diversity, stand age, site productivity or stand density are the main drivers to influence variations in C stocks across secondary subtropical forests. The effects of stand age on C stocks may be direct (Becknell and Powers, 2014) or indirect (i.e., mediated through stand characteristics) on forest C. For example, stand age leads to changes in the composition of plant species over the course of succession, by which shade-intolerant

species are replaced with shade-tolerant species (Vayreda et al., 2012). We predicted that C stocks would increase with stand age, but after accounting for stand age, residual variations could be explained by a combination of species diversity, DBH diversity, height diversity, site productivity, and stand density (Fig. 1). Thus, stand age may be the primary driver of C stocks in secondary subtropical forests, as previous works have suggested that stand age is a

strong determinant of stand growth (Powers et al., 2009; Becknell and Powers, 2014).

To test our hypothesized relationships between stand age, stand characteristics, and C stocks across subtropical forests, we randomly selected 80 forest plots with different stand ages in Eastern China. Specifically, we asked the following questions in accordance to relative contribution of stand characteristics for explaining variations in C stocks: 1) are stand

structural diversity, stand age, stem density, and site productivity associated with

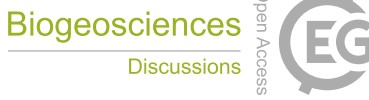



aboveground C and soil organic C stocks? 2) what are the relative contributions of stand

structural diversity *versus* stand age, stand density, and site productivity to variations in

aboveground C and soil organic C stocks in subtropical forests? and, 3) what are the direct

and indirect effects of stand age on variations in aboveground C and soil organic C in these

subtropical secondary forests?

## 2      Materials and methods

### 2.1      Study site, plots and forest structure

The study was conducted in the lower eastern extension of the Tiantai and Siming Mountains

(29º41-50´N, 121º36-52´E) located near Ningbo City, Zhejiang Province, in Eastern China.

This region has a typical subtropical monsoon climate with a hot and humid summer and a

dry cold winter. The highest peak in this area reaches 800 m above sea level, while most

other reliefs are in the 70-500 m range (Song and Wang, 1995). The studied region had been

subject to different intensities of human disturbances (typically logging), but have been

protected from this activity for the last 25 years. Consequently, forests in the region contained

stands at different developmental stages (Wang et al., 2007; Yan et al., 2009).

We randomly sampled the stands in the area that meet the criteria—naturally recovered

stands (no recent disturbances). The soils in these areas were classified as Ferralsols

according to the FAO soil classification system (WRB, 2006), which is equivalent to the

Yellow or Red Soils in the Chinese soil classification system, with the parent materials

consisting mostly of Mesozoic sedimentary rocks, some acidic igneous rocks, and granite

residual weathered material (Song and Wang, 1995). We established a total of 80 plots,

covering all typical habitats in this region. Each plot (20 × 20 m) was located at a distance of

least 100 m from stand edges in order to minimize edge effects. We acknowledge that our

plot sizes were quite small; however, similar to other regions, secondary forest patches often



occur in smaller tracts than is the case with primary forests (Becknell and Powers, 2014). A description of the vegetation and soil characteristics is provided in Table 1.

In each plot, the basal diameter (diameter at 5 cm above root collar) and DBH were measured for trees taller than 1.50 m, while the basal diameter and diameter at 45 cm above the ground ($D_{45}$) were measured (with a diameter tape) for trees that were shorter than 1.50 m. Total tree height for each tree was measured with a telescopic pole for the height of up to 15 m, and with a clinometer for heights of >15 m. The studied plots had between six and 46 tree species per plot, and among them, deciduous species such as *Liquidambar formosana* and *Quercus fabri*, and evergreen species such as *Lithocarpus glaber* were the dominant species in young forests, with evergreen species such as *Choerospondias axillaris* and *Schima superba* dominating in the premature forests, while *Castanopsis fargesii* and *Castanopsis carlesii* dominated in the mature forests.

### 2.2    Estimations of stand age and site productivity

Stand age represents the number of years since the stand replacing disturbance (e.g., Wang et al., 2007; Yan et al., 2009). The official documents of Ningbo Forestry Bureau, Zhejiang Province, were reviewed to collect relevant data about the disturbances in the study area.

Previous work has shown that community vertical structures and plant species compositions were similar within each forest developmental stage in our study area (Yan et al., 2013). Thus, we assessed site productivity for each studied plot through direct volume measurements using a dendrometric (phytocentric) method (Skovsgaard and Vanclay, 2008). Site productivity was calculated as the mean annual increment of stand volume per year based on stand volume per hectare (Loetsch and Haller, 1964) divided by stand age, which represents productivity accumulated from stand establishment (e.g., Pretzsch et al., 2014).



It was to note that tree diameter of each individual was used for calculating individual

aboveground biomass (AGB; Brown et al., 1989; Ali et al., 2015), and hence tree AGB scales

closely with the volume of the individual tree ($R^2 = 0.93$; $P < 0.001$ in this study). This is

somewhat different from stand volume per hectare (Loetsch and Haller, 1964; Pretzsch et al.,

2014). A high stand volume per hectare can be caused by many small trees (each containing

little AGB) and/or a few big trees (each containing a disproportionately large AGB; e.g.,

Liang et al., 2007; Lei et al., 2009; Wang et al., 2011; Slik etal. 2013; Poorter et al. 2015; see

Fig. S1). In addition, stand basal area per hectare (used in the calculation of stand volume)

has been proved as a useful proxy of productivity in secondary subtropical forests of China

(e.g., Barrufol et al., 2013).


### 2.3    Measurements and calculations of carbon stocks

For individual trees with DBH ≥5 cm, aboveground biomass (AGB$t$) was calculated using the

Brown's allometric equation (eqn 1; Brown et al., 1989) with DBH only because tree height

and DBH of the studied subtropical trees was highly correlated ($r = 0.86$, $P < 0.001$).

190                $AGBt = exp\{-2.134 + 2.530 \times Ln(\text{D})\}$          eqn 1

where D is diameter at breast height.

     To avoid the uncertainty about using of Brown's equation for our studied forests, we

have developed regression relationship between basal area (substitute of AGB) and DBH (≥

5cm) for the species in our studied system. It is found that the Brown's equation and our

developed regression equation, for basal area – DBH, yielded almost similar relationships

(Fig. S1). In addition, previous work has shown that basal area was highly related with AGB

(Ali et al. 2014), and the D-H models for AGB could be generally used across subtropical

large trees, small trees and shrubs (Ali et al. 2015). Further, Brown's equations had





commonly used for estimation of AGB in different subtropical forests (e.g., Conti and Díaz,

200    2013).

In addition, the individual tree AGB (DBH $\geq$ 5 cm) estimated with Brown's (1989)

equation was compared with each of simple geometric equation and most recent equations

using plant height and wood density (such as Chave et al.'s 2005 equations; see Fig. S3). We

found that the Brown's equation tended to over-estimate individual tree aboveground

biomass as compared to the estimations obtained using simple geometrical equation, but the

results of the two models were highly consistent ($R^2$ = 0.91, P < 0.001; see Fig. S3a). Further,

the Brown's equation also tended to over-estimate individual tree aboveground biomass as

compared to the estimations obtained using Chave et al.'s (2005) $\rho D^2 H$ model while almost

similar estimations to Chave's $\rho D$ model for moist forests, but the results of the models were

highly consistent ($R^2$ = 0.91 and 0.96 with P < 0.001 for two equations of Chave's with

Brown's equation; see Fig. S3b and c). These results were therefore consistent with recent

continental scale study (Paul et al., 2016) showing that when comparing the estimated AGB

through model using stem diameter as a single predictor there was little improvement in

accuracy of estimation when the model included other plant variables (e.g. height, wood

density).

We estimated AGB of individual shrubs and small trees (AGB$s$) using a diameter-height

(DBH < 5 cm) based multi-species equation developed locally ($n$ = 96, $R^2$ = 0.71, $P$ < 0.001;

Ali et al., 2015).

$$AGBs = 1.423 \times \exp\{-3.50 + 1.65 \times Ln(D) + 0.842 \times Ln(H)\} \qquad eqn\ 2$$

where D is DBH < 5 cm, and H is tree height (m).

The sum of the aboveground biomass for trees and shrubs was considered as total AGB

per plot. Subsequently, we converted AGB to aboveground C stock (Mg ha$^{-1}$) by multiplying

AGB with a factor of 0.5, as 50% of the total tree biomass being C (Dixon et al., 1994).



Soil samples were collected from 0–20 cm depth from 65 sample plots. Soil samples in

each plot were collected from five randomly selected points, resulting in 325 samples, which

were taken to the laboratory and air-dried over 30 days. Each soil sample was then sifted

through a 0.25 mm sieve and thoroughly mixed to determine organic soil C concentrations

using the oil bath-$K_2CrO_7$ titration method (Nelson and Sommers, 1974). In each plot, soil

bulk density was determined using a steel corer of a known volume, and five soil cores were

collected per plot. The soil cores were dried in at 105 °C in an oven for > 48 hours, after

coarse fragment such as stone was removed. Bulk density (g cm$^{-3}$) was calculated by dividing

the oven dry weight of the soil (g) by the volume of the soil core. The amount of soil organic

C (Mg ha$^{-1}$) was calculated by multiplying the organic C content by the soil depth and soil

bulk density (Brown, 2004).


### 2.4     Calculation of stand structural diversity

We selected the Shannon-Wiener biodiversity index to quantify tree size variation (Magurran,

2004). With the Shannon–Wiener index, DBH and height were grouped into discrete classes.

For DBH, 2, 4, 6, and 8 cm classes were tested, while for height, 2, 3, 4, and 5 m classes were

tested in order to calculate the indices. We assessed the correlation between DBH diversity

and height diversity with different classes of DBH and height, respectively, for the purpose of

stand structural management (e.g., Lei et al., 2009). Hence, the highest correlation coefficient

($r = 0.54$, $P < 0.001$) between DBH diversity and height diversity was achieved with DBH

and height classes of 8 cm and 3 m increments, respectively. Therefore, 8 cm and 3 m

increments were utilized for the DBH and height classes in calculating DBH and height

diversity, respectively. Based on basal area proportions, tree species, DBH and height

diversities were calculated using equations 3for each plot (Buongiorno et al., 1994;

Staudhammer and LeMay, 2001; Magurran, 2004).



$$H_x = -\sum_{i=1}^{x} p_x \times Log\, p_x \qquad \text{eqn 3}$$

where $H_x$ was either species diversity, DBH diversity or height diversity; $p_{xx}$ was either the proportion of basal areas of $x$th species, $x$th diameter classes or $x$th height classes, respectively, while $x$ was either the number of tree species, diameter or height classes, respectively.

The analysis on the Shannon-Weiner indices was performed using the *vegan* package for the R (Oksanen et al., 2015; R Development Core Team, 2015).

### 2.5    Statistical analysis

We conducted three sets of data analysis. Firstly, we used a simple linear regression analysis to test for pair-wise associations of C stocks (aboveground and/or soil organic) with each of species diversity, DBH diversity, height diversity, stand age, stand density, and site productivity. We also tested the pair-wise association between stand age and species diversity, DBH diversity, height diversity, stand density, and site productivity.

Secondly, three series of ordinary least squares (OLS) multiple regressions analyses were conducted to test whether C stocks (aboveground and/or soil organic) were primarily driven by stand structural diversity (species, DBH, and height diversity; first series), other characteristics of the stand (stand age, stand density, and site productivity; second series), and a combination of stand structural diversity and other stand characteristics (third series). The OLS multiple regression analyses were conducted using the Spatial Analysis in Macroecology software package (SAM version 4.0; Rangel et al., 2010). Regressions were developed for each C stock response variable by starting from three potential predictor variables (species diversity, DBH diversity, and height diversity; or stand age, stand density and site productivity) without interactions, resulting in a total of seven possible models for each of the first and second series (Fig. 1). With respect to the third series for each response



variable, a total of 63 possible models were tested by beginning from six potential predictor variables (species diversity, DBH diversity, height diversity, stand age, stand density, and site productivity; Fig. 1). For the significance test, the model with the lowest Akaike Information Criterion (AICc; Akaike, 1973) was selected as being the best for each series. In addition, a model averaging approach (synthetic model) was developed in SAM to evaluate which predictor variable contributed consistently across all the models of each series. For this, regression coefficients of each predictor were averaged across all models of each series, and weighted by their Akaike Information Criterion weight (AICc-wi), which represented the likelihood of a given model relative to all other models (Wagenmakers and Farrell, 2004). An importance value was calculated by adding the AICc-wi values of the models in which the variables were present (Slik et al., 2013). Importance values ranged between zero (low importance) and one (high importance). For each response variable, the final best model among the three competing series was selected on the basis of the lowest AICc. It is worth mentioning here that aboveground C stock, DBH diversity and site productivity were calculated using tree diameters, thus, we ran the multicollinearity statistics. Multicollinearity diagnosis was performed in multiple regressions using the variance inflation factor (VIF) as multicollinearity larger than 10 could cause inaccurate model parameterization and decreased statistical power, and exclude significant predictor variables (Graham, 2003).

Lastly, we employed a structural equation model (SEM) to assess the direct effects of species diversity, DBH diversity, height diversity, stand age, stand density, and site productivity on C stocks (aboveground and/or soil organic), and the indirect effects of stand age, on each of the C stocks through the mediation of other stand characteristics. However, even if VIF value is lower than 10, it may still cause inaccurate model parameterization, decrease statistical power and exclude significant predictor variables. Hence, it potentially impairs the identification of significant effects and invalidates approaches that assume no





collinearity among predictor variables (Graham, 2003). Thus, several tests in SEM were used

to assess model fit, i.e., the Chi-square test, goodness of fit index (GFI), comparative fit index

(CFI), minimum discrepancy (CMIN/df), Root mean square error of approximation

(RMSEA) and Akaike Information Criterion (AIC). The SEM is an advanced and robust

multivariate statistical method that allows for hypotheses testing of complex path-relation

networks (Malaeb et al., 2000); assuming linear relationships and correlations between

variables in the model. Here, we tested two different models, a stand characteristics model

and a stand age model (Fig. 1). The stand characteristics model was the best finally selected

model among the three competing OLS series (see second step of the statistical analysis).

Thus, we tested the direct effects of the stand characteristics, and retained predictor variables

in the final best model on C stocks. With respect to the stand age model, stand age was

employed as the primary explanatory variable by testing the direct and indirect effects

(mediated by stand characteristics) on C stocks. The SEM analyses were conducted using

IBM SPSS Amos (version 21), and a summary of variables and their categories are described

in Table S1.

It is to note that the largest (dominant) trees could also determine the total number of

diameter classes (e.g. "size richness" based on Eqn. 3). Therefore, it is necessary to justify

whether significant effects of stand structural diversity on C stocks in the regressions

and/SEM are caused by the "diversity" of tree structure frequency distribution, rather than by

the dominant characteristics of trees. As such, we conducted a Pearson correlation analysis on

the relationships between stand structural diversity (i.e., tree DBH and height) and each of

90-percentile diameter/height (i.e., P90 of D/H) and coefficient of variation in

diameter/height (i.e., CV of D/H). If the proposed stand structural diversity indices were

more related to the CV of D/H, the significant results in the regressions and/or SEM on a

response variable would be caused by the "diversity" of tree structure frequency distribution,



rather than by the characteristics of dominant trees in forests. We found that tree DBH and

height diversity indices had significantly stronger relationships with tree structure frequency

distribution (e.g., CV of D and H) than with the dominant characteristics of trees in forests

(e.g., P90 of D/H, Table S8).

## 3 Results

### 330 3.1 Relationships between stand characteristics and carbon stocks

Aboveground C stock was positively related to tree DBH diversity (Fig. 2a), tree height

diversity (Fig. 2b), and site productivity (Fig. 2c), which explained 49, 13, and 41 % of the

variation, respectively. There was no significant relationship between aboveground C stock

and species diversity and density (Table S2). Soil organic C stock was not significantly

related to stand age or other stand characteristics (Table S2).

Stand age was positively related to aboveground C stock, and explained 50 % of the

variation in aboveground C stock (Fig. 2d). Mature stands exhibited a greater range in tree

DBH and height distribution in that they had a greater number of large trees overall (Fig. S4a

and b). Aboveground C stock was observed to ranged widely across forests, from 3.15 to

238.91 Mg ha$^{-1}$, and forests with similar ages had different levels of aboveground C stock

(Fig. 2d).  Stand age also explained 39 and 30% of the positive variation in each of tree DBH

(Fig. 2e) and height diversities (Fig. 2f). However, stand age did not explain any of the

variation (≤ 2%) in species diversity, site productivity, and stand density (Table S2).

### 345 3.2 Relative contribution of stand characteristics to carbon stocks

When testing the effects of species, DBH, and height diversities on aboveground C stock

(first series) by using the best regression model ($R^2 = 0.54$, $P < 0.001$), we found that

aboveground C stock was negatively related to species diversity, but positively related to





DBH diversity (Table 2). Further, in the synthetic model, the significant predictors with the

highest importance values were DBH diversity (1.0) and species diversity (0.97; Table S3). In

contrast, tree height diversity was not significant in both the synthetic and the best models.

For the testing of the second series, aboveground C stock was positively correlated to stand

age and site productivity, but negatively related to stand density in the best regression model

($R^2 = 0.79$, $P = 0.001$; Table 2). In the synthetic model, all three predictors were significant;

however, stand age and site productivity had the similar highest importance value (1.0) as

compared to stand density (0.70) (Table S3). When species diversity, DBH diversity, height

diversity, stand age, stand density, and site productivity were jointly tested (third series), the

best regression model ($R^2 = 0.83$, $P < 0.001$) revealed that aboveground C stock was

positively correlated to stand age, site productivity and DBH diversity, but negatively related

to species and height diversity (Table 2). In the synthetic model, the significant predictors

with high importance value were stand age (1.0), site productivity (1.0), species diversity

(0.96), DBH diversity (0.90) and height diversity (0.77; Table S3); however, stand density

was not significant in both the synthetic and the best models. It is worthy of mention that the

best model of the third series was the best-fit model among the competing best models of all

three series, in that it had the lowest AICc as well as the highest $R^2$ (Table 2).

With respect to organic soil C stock, the best models of all series revealed that none of the

species diversity, DBH diversity, height diversity, stand age, stand density, and site

productivity had significant effects (Table S4). Although some of the predictors were retained

in the best models of each series, they were not significant and explained very low variations

in soil organic C stock ($R^2$ values ranged between 0.00 and 0.03; Table S4). Also, in the

synthetic model of each series, the importance values of the predictor variables were very low

within the range of 0.29-0.50 (Table S5). It was noted that all VIF values were lower than the



critical heuristic value of 10, which suggested that collinearity among predictor variables did

not strongly affect our results (Table S6).


### 3.3 Direct and indirect effects of stand age on carbon stocks

Stand characteristics models and stand age models yielded almost identical fit measures (Chi-

square = 4.48 and 2.24, *df* = 5 and 3, P-value = 0.483 and 0.486, CFI = 1.00 and 1.00, GFI =

0.98 and 0.99, CMIN/df = 0.90 and 0.81, RMSEA < 0.001 and < 0.001, respectively (Fig. 3).

The stand characteristics model explained 81% of the variation in aboveground C stock (Fig.

3a), while the stand age model explained 83% (Fig. 3b).

In the stand characteristics model, aboveground C stock was directly linked with stand

age, species diversity, DBH diversity, height diversity, and site productivity (Fig. 4a).

According to the final best model in OLS series (Table 2), and in order to achieve the best-fit

model in SEM, the non-significant relationship between aboveground C stock and stand

density was removed (Fig. 3a). Thus, the size (standardized regression weight: beta) of the

direct effects of stand age, species diversity, DBH diversity, height diversity, and site

productivity on aboveground C stock was 0.61 ($P < 0.001$), -0.19 ($P < 0.001$), 0.24 ($P =$

0.001), -0.14 ($P < 0.028$), and 0.46 ($P < 0.001$), respectively (Fig. 4a; Table S7). In the stand

age model, 39% and 30% of the variations in DBH diversity and height diversity were

explained by stand age (Fig. 3b). In contrast, stand age did not explain the variations (< 2%)

in species diversity, site productivity, and stand density (Fig. 3b). Considering the total

effects of stand age (sum of direct and indirect effects), aboveground C stock was positively

affected by the sum of the direct (positive) and indirect (positive) effects of stand age through

species diversity (negative), DBH diversity (positive), height diversity (negative), and site

productivity (positive) (Fig. 3b). Aboveground C stock was not indirectly affected via stand

density by stand age (Fig. 3b). Although the effect of stand age on aboveground C stock was



stronger directly (beta = 0.59) than indirectly (beta = 0.11), the total effect of stand age was
significant and stronger, with an effect size of 0.70 ($P < 0.001$; Table S7).

For soil organic C stock, the stand characteristics model revealed that the direct
relationships between each of the stand characteristics and soil organic C stock was not
significant (Fig. S5a). Also, in the stand age model, the direct and indirect effects of stand age
on soil organic C stock were not significant (Fig. S5b).

**4     Discussion**

The significant relationships of stand characteristics with aboveground C stock, but not with
soil organic C stock, in the studied forests suggest that, relative to soil organic C stock,
aboveground C stock is more predictable with respect to aboveground stand attributes. It is
understandable that stand characteristics were derived from the aboveground forest structure.

It may be the case that soil organic C stock is related to belowground stand characteristics,
which were not studied in this research.

**4.1 Relationship between stand structural diversity and aboveground C stock**

Our results indicated that tree DBH diversity and height diversity were positively correlated

with aboveground C stock across plots, indicating that stand structural diversity is one of the
key factors that affect aboveground C stock in subtropical forests (Fig. 2). The strong positive
relationships between aboveground C stock and stand structural diversity might result from
high resource use efficiencies initiated by complex tree size structures (Vayreda et al., 2012).
Tree species possessing different diameters and heights may have their own set of habitat

requirements for nutrients and coverage (Wang et al., 2011). The maintenance of high stand
structural diversity support species to meet their specific requirements, whereas low or
homogenous structural arrangements may reduce complementarity effects (Lei et al., 2009).



Therefore, significant variations in tree DBH and heights may result in a multilayered forest

structure with enhanced structural complexity, allowing for more efficient light capture at the

stand level, leading to a larger accumulation of aboveground C stock (Buongiorno et al.,

1994; Staudhammer and LeMay 2001; Zhang and Chen 2015).

It is worth noting that tree species diversity had a non-significant and negative pair-wise

association with aboveground C stock (Table S2), which likely resulted from increased

species richness, while species evenness decreased through stand development in the forests

under study (Table 1). Although biomass should increase with species richness and evenness

(e.g., Zhang et al., 2012), the explanation for why we did not observe a positive effect of

species diversity on aboveground C stock might be that less diverse stands were dominated

by more productive species, such as those that are early successional. Furthermore, tree

species diversity decreased slightly from young to premature stands, which leveled to

constant, from premature to mature stages (Table 1). This might result in a weak relationship

between species diversity and aboveground C stock during forest stand development. In the

SEM analysis, however, we found negative relationships between aboveground C stock with

species and height diversities (Fig. 3), likely stemming from the complex shift patterns of

species diversity through forest succession, as discussed above, which was also observed in a

semi-deciduous tropical forests (Larpkern et al., 2011). In addition, we also included the

effects of other stand characteristics in the SEM analysis, but did not consider the effects of

other factors on aboveground C stock in the simple linear regression. In this situation, the

relationship of species diversity with aboveground C stock includes the combined effects of

other stand characteristics on aboveground C stock in the SEM analysis.

Similarly, the negative relationship between aboveground C stock and tree height

diversity was observed in the SEM analysis. However, we found that the relationship

between tree height diversity and aboveground C stock was positive in the simple linear



regression. These contrasting results suggest that the association between height diversity and

aboveground C stock is uncertain, and largely contingent on whether additional effects of

other stand characteristics on aboveground C stock are considered. When the effects of other

stand characteristics were considered in the SEM analysis, there was a negative effect of tree

height diversity on aboveground C stock. The negative relationship of aboveground C stock

with tree height diversity in the SEM model demonstrated that forest stands with high tree

height diversity may reduce aboveground C stock through the alternation of other stand

characteristics, such as shifting species composition during forest succession. Forest stands

with high tree height diversity, but without high tree DBH diversity and increasing stand age,

may have low aboveground C stock. Generally, aboveground C stock might be more loosely

correlated to tree height alone, but is likely correlated with the combination of the tree height

and the growth rates of tree species. For instance, some of the most extensive aboveground C

stock observed in the old growth conifer forests, were associated with the slow growth of tree

species (e.g., Gahagan et al., 2015). Conversely, shrublands and young forests dominated by

deciduous species with very high growth rates were associated with low aboveground C stock

in the study area (Yan et al., 2013). Therefore, it was clear that, rather than great height

diversity, tree species with low height diversity and great DBH diversity maintained high

aboveground C stock in the forests under study.

## 4.2    Stand structural diversity and site productivity mediate the relationship between stand age and aboveground C stock

In this investigation, stand structural diversity, site productivity, and stand age, in

conjunction, explained more variation in aboveground C stock than did singular components,

such as stand structural diversity or other stand characteristics. More importantly, stand

characteristic models and stand age models provided strong support for our prediction that



stand age, site productivity, and stand structural diversity could jointly explain large

variations (i.e., 81%) in aboveground C stock. Therefore, our hypothesis was partially

confirmed, i.e., stand structural diversity, stand age, or site productivity alone, or jointly,

comprised the drivers of variations in C stocks across the forests under study.

The clearly positive contribution of stand age and site productivity to aboveground C

stock might relate to successional patterns of tree growth and other stand characteristics. This

study revealed that, relative to other stand characteristics, stand age was the most significant

factor in predicting aboveground C stock (Table 2; Fig. 2d; Fig. 3a), which were also found

in tropical dry and seasonal forests (Dupuy et al., 2012; Becknell and Powers 2014).  It is true

that stand age may affect aboveground C stock directly, as tree DBH increases when forest

stands become mature (Lohbeck et al., 2015). In general, mature stands typically contain old

large trees. The AGB growth rates (C stock accumulation rates), for tree species increases

with tree DBH (Slik et al., 2013; Stephenson et al., 2014). A set of large trees in mature

stands may add the same level of C to the forest within a year as do all of the mid-sized trees

contained in the same forest (Stephenson et al., 2014). In this study, we found that variation

in aboveground C stock was mainly affected by tree structure frequency distribution (e.g., CV

of D and H), compared to the dominant characteristics of trees in forests (Table S8; Zhang

and Chen, 2015). Consequently, a positive relationship must exist between aboveground C

stock and stand age. In this case, stand age acts as a primary determinant of stand growth

(Powers et al., 2009; Becknell and Powers, 2014), to drive variation in aboveground C stocks

(e.g., Chen and Luo, 2015).

Stand age might also indirectly impact aboveground C stock through the directional

changes in other stand characteristics during forest succession (Campetella et al., 2011;

Lohbeck et al., 2015). As expected, we found that stand age was significantly related to tree

DBH and height diversity (Figs 2 and 3b), which had a significant influence on aboveground




C stock (Fig. 3b). The positive contribution of site productivity to aboveground C stock

during stand development was also found in secondary tropical forests (Lohbeck et al., 2015).

It is well known that increases in forest productivity and biomass play a critical role in

shaping C accumulation through high nutrient supply (Giardina et al., 2003). In this study,

most of the stands were still recovering from disturbances, thus site productivity and nitrogen

availability increased with stand development (Yan et al., 2009). As a result, aboveground C

accumulation increased through forest succession.

Distinguishing the direct and indirect effects of stand age through mediations of stand

characteristics on aboveground C stock may determine the role that stand age plays in driving

variation in C stock during forest succession. By employing a structural equation model, we

observed that stand age could explain a small additional variation (~ 2%) in aboveground C

stock when it was considered as a primary driver of aboveground C stock through the

mediation of stand characteristics (Fig. 3b). However, the results showed that stand age had

substantial direct and total effects (sum of direct and indirect effects) on aboveground C stock

(Fig. 3b). Clearly, these contrasting results indicated that the direct effects of stand age on

aboveground C stock was much stronger than the indirect effects of stand age through the

mediation of stand characteristics in the forests under study. The possible reasons for the low

indirect effects of stand age on aboveground C stock in this investigation might be attributed

to the contributions of the other factors such as environmental properties and species

competition, which were not included in our model.  It should be noted that this study did not

focus on the association of C stock with environmental properties, or tree mortality rates,

recruitment, and survival. However, these biotic and abiotic factors also have linkages with

stand age toward the influence of C stock in forest ecosystems (Giardina et al., 2003; Lutz

and Halpern, 2006; Liang et al., 2007; Lei et al., 2009; Vayreda et al., 2012; Chen and Luo,

2015). Therefore, we suggest that further research should be conducted to improve our model



by including the direct and indirect effects of environmental properties, as well as the

demographic traits of tree species on the relationship between stand age and C stock.


## 5    Conclusions

This study has presented and articulated the inherent complexities of variation, as relates to

aboveground C stock, by utilizing six stand characteristics of secondary subtropical forests

across Eastern China. We found that 81 % of variations in aboveground C stock could be

explained by stand characteristics in these heterogeneously aged forests (Fig. 3a). However, it

is noteworthy here that stand age is the main driver directly and indirectly, via stand

structural diversity and site productivity, affecting variation in aboveground C stock in

subtropical secondary forests (Fig. 3b). Rather than species and height diversities, DBH

diversity, stand age and site productivity cumulatively contributed to variation in

aboveground C stock during stand development in subtropical forests in Eastern China.

Therefore, improving tree DBH diversity and stand condition could be an effective approach

for continue C storage in subtropical forests.

## Acknowledgements


This work was supported by the National Natural Science Foundation of China (Grant No.

31228004 and 31270475). We thank Professor Scott X. Chang (University of Alberta) and

Professor Xuhui Zhou (East China Normal University) for their comments on early draft.

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



**Table 1** Characteristics of three forest development stages considered for the study on the linkage between stand characteristics and carbon stock (vegetation and soil) in subtropical evergreen broadleaved forests in Eastern China. Values are mean $\pm$ *SD* for each development stage. Values with different lowercase letters in a given row are significantly different at *P* < 0.05 (*LSD* Fisher). The number of plots used (*n*) for young forests, premature forests and mature forests was 21, 39, and 20, respectively, for the vegetation data, and 21, 25, and 19, respectively, for the soils data.

| | Young forest | Pre-mature forest | Mature forest |
|---|---|---|---|
| ***Vegetation structure*** | | | |
| Maximum tree height (m) | 14.2±7.5a | 21.8±5.1b | 24.1±5.1b |
| Maximum tree DBH (cm) | 19.2±7.3a | 38.6±11.5b | 47.7±11.2c |
| Species richness | 21±9a | 26±10ab | 29±8b |
| Species evenness | 0.6±0.1a | 0.6±0.2a | 0.5±0.2a |
| Tree biomass (Mg ha$^{-1}$) | 48.2±31.3a | 172.7±76.5b | 256.81±105.8c |
| aboveground C stock (Mg ha$^{-1}$) | 24.1±15.7a | 86.4±38.2b | 128.4±52.9c |
| Tree species diversity index | 2±0.6a | 2±0.5a | 2±0.5a |
| Tree DBH diversity index | 1±0.3a | 1±0.3b | 2±0.3b |
| Tree height diversity index | 1±0.5a | 2±0.2b | 2±0.2b |
| Age (year) | 22±4.7a | 79±6.1b | 125±6.9c |
| Site productivity (m$^3$ ha$^{-1}$ year$^{-1}$) | 3.3±2.1a | 4.4±2.2a | 3.7±1.7a |
| Stand density (stems ha$^{-1}$) | 6068±3371a | 4970±2457a | 4512±1857a |
| ***Soil property (0-20 cm)*** | | | |
| Bulk density (g cm$^{-3}$) | 1.2±0.2a | 1.1±0.2ab | 1±0.2b |
| Soil organic C stock (Mg ha$^{-1}$) | 80.8±26.7a | 85.3±28.8a | 87.3±20.6a |
| ***Forest management history and land-use regime*** (Yan et al., 2009) | Naturally regenerated stands after harvesting. In recent decades, forest harvesting has declined due to the availability of natural gas for cooking and heating. | Snags and downed deadwood harvesting. Nature disturbance regimes including typhoon and landslide. | Protected from clear-cutting. The stands were in the canopy gap-phase. Typhoon is the major disturbance regime (that returns 1-3 years) at the regional scale. |

DBH: Diameter at breast height



**Table 2** The best model obtained from a series of regression analyses of a response variable (aboveground C stock) on stand structural diversity (species, DBH, and height diversities; first series), other stand characteristics (stand sage, stand density, and site productivity; second series), and a combination of stand structural diversity and other stand characteristics (third series). For each predictor variable, the regression coefficient (Coeff.), standardized regression coefficient (Beta), $t$-test and $P$-value are given. The coefficient of determination ($R^2$), $F$-test, $P$-value and Akaike Information Criterion (AICc) of the model are also given. For each effect of the first and second series, all seven possible models were tested, while all 63 possible models were tested for the third series. See Table S3 for the contribution to the models of all variables tested. Detailed statistics of all models for the first, second, and third series are provided in Tables S9, S10, and S11, respectively. $P$ values < 0.05 are given in bold.

| Model and predictor variable | Coeff. | Beta | $t$ | $P$ | $R^2$ | AICc |
|---|---|---|---|---|---|---|
| **Effects of stand structural diversity** | | | | | | |
| *Model*[1] | | | 45.68 | **<0.001** | 0.54 | 809.22 |
| Constant | -15.38 | 0.00 | -0.82 | 0.414 | | |
| Species diversity | -23.36 | -0.24 | -3.02 | **0.003** | | |
| DBH diversity | 105.17 | 0.74 | 9.47 | **<0.001** | | |
| **Effects of other stand characteristics** | | | | | | |
| *Model* | | | 97.69 | **<0.001** | 0.79 | 747.66 |
| Constant | -26.85 | 0.00 | -3.02 | **0.003** | | |
| Stand age | 0.81 | 0.61 | 11.40 | **<0.001** | | |
| Site productivity | 14.72 | 0.57 | 10.44 | **<0.001** | | |
| Stand density | -<0.01 | -0.11 | -1.98 | **0.05** | | |
| **Joint effect of stand structural diversity and other characteristics** | | | | | | |
| *Model* | | | 70.06 | **<0.001** | 0.83 | 739.12 |
| Constant | -5.16 | 0.00 | -0.34 | 0.736 | | |



| | | | | |
|---|---|---|---|---|
| Species diversity | -17.42 | -0.18 | -3.46 | **0.001** |
| Height diversity | -19.93 | -0.13 | -2.12 | **0.037** |
| DBH diversity | 33.70 | 0.24 | 3.08 | **0.003** |
| Site productivity | 11.33 | 0.44 | 7.66 | **<0.001** |
| Stand age | 0.77 | 0.58 | 8.74 | **<0.001** |

[1] The value under $t$ column represents $F$-test of the model



## Figure Legends

**Fig. 1** Conceptual model for explaining C stocks in secondary subtropical forests in Eastern

China. The general model represented two basic models. 1) Model of the direct

effects of stand structural diversity (i.e., species diversity, DBH diversity, height

diversity) and other stand characteristics (i.e., stand age, stand density, and site

productivity) on C stocks (stand characteristics model; indicated by black solid

arrows). 2) Model of the direct and indirect effects of stand age through mediations of

the stand structural diversity and other stand characteristics (stand age model;

indicated by gray dashed arrows). Note that the one-sided solid or dashed arrow with

black or gray color represents regression path, and the two-sided arrow with black

color represents correlation between variables.

**Fig. 2** Relationships between stand characteristics and C stocks and between stand age and

stand structural diversity in subtropical evergreen broadleaved forests. Only

significant associations (see Table S2) are shown here **(a-d)** Aboveground C stock

(ACS) as a function of tree DBH diversity, tree height diversity, site productivity,

stand age; **(e)** DBH diversity as a function of stand age; and **(f)** height diversity as a

function of stand age.

**Fig. 3** Best-fit structural equation models for aboveground C stock; a) combining species

diversity, DBH diversity, height diversity, stand age, stand density, and site

productivity (stand characteristics model), b) stand age as a primary explanatory

variable by testing direct and indirect effects through mediation of stand

characteristics ( stand age model) across all 80 subtropical forest plots. Stand

characteristics model includes correlations among species diversity, DBH diversity,





height diversity, stand age, stand density, and site productivity. However, the selected

best model of the third series excludes these correlations (see Table 2). Values give

the standardized coefficients for the correlation between variables; all coefficients are

significant at *, $P < 0.05$; **, $P < 0.01$; ***, $P < 0.001$; ns, non-significant; and

coefficient of determination ($R^2$) for response variables are indicated. Epsilons (ε)

within small circles represent the error term for downstream variables, ellipses

represent response variable (aboveground C stock), and squares or rectangles

represent predictor variables. But in the case of model (b), the squares or rectangles

with white fill represent mediators, while those with gray fill represent primary

variables.



**Fig. 1**

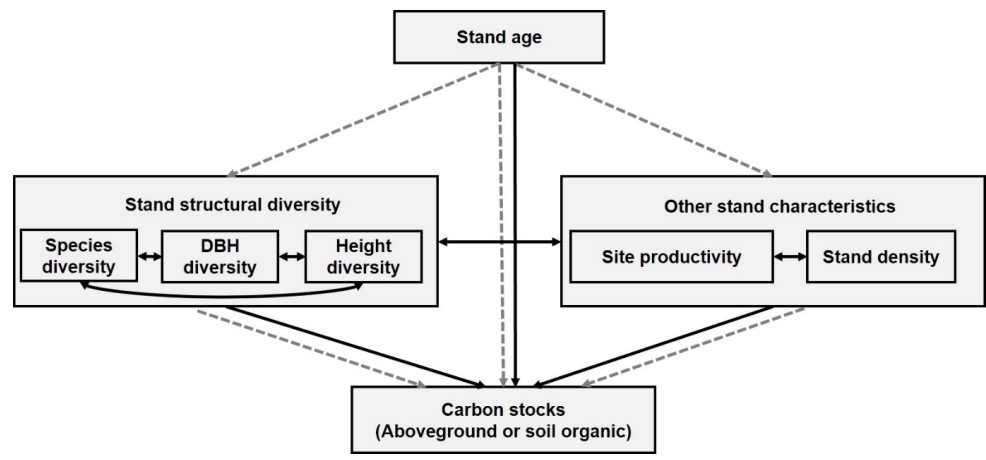





**Fig. 2**

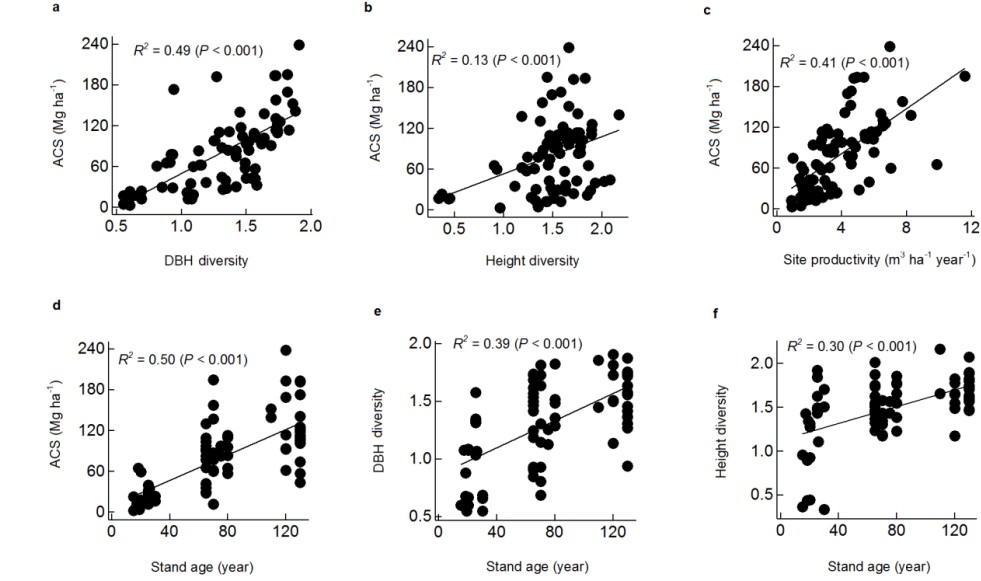






**Fig. 3**


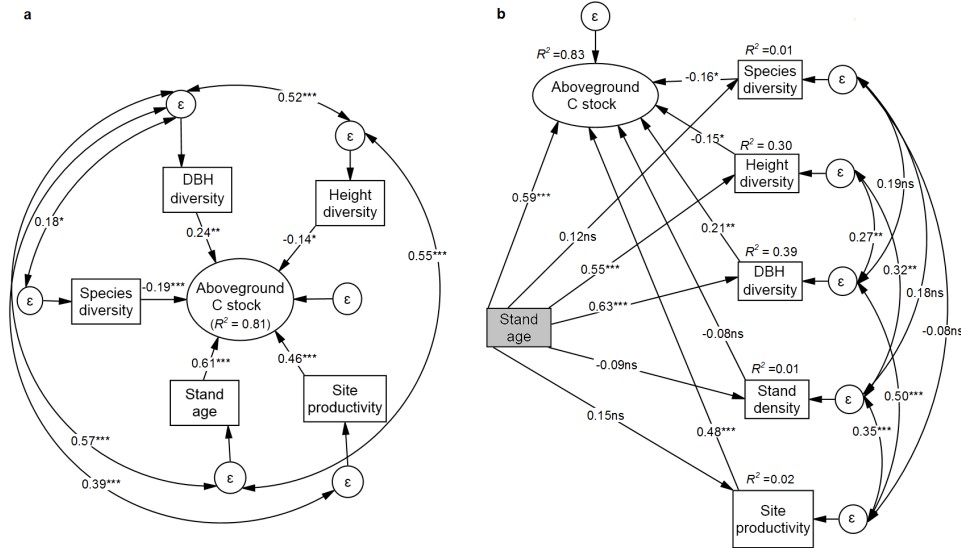