# Peer review of "Relative contribution of stand characteristics on carbon stocks in subtropical secondary forests in Eastern China"

_Biogeosciences, 2016_

## Referee Comment (RC1) · Anonymous Referee #1 · 17 Mar 2016

Ali et al. present a study on an interesting and important topic: biomass estimation for subtropical forests in the East Asian monsoon region. The study is generally well introduced and clearly structured. The data set is most probably appropriate to tackle the research questions raised by the authors. The choice of analytical methods, however, needs considerable reconsideration in some regards.

1) measurements and calculations of carbon stocks

- There are no measurements of carbon stocks, just calculations based on allometric equations, so please adjust the section title accordingly.

- I was not able to find eqn 1 in Brown et al. 1989, please indicate exact reference or

modification if applicable.

- 14% of variance in tree height are not explained by diameter. This information could be used to improve allometric estimates, since the diameter-height-allometry varies with environmental conditions, and might provide valuable additional information.

- However, there is no way of validating your AGB estimates, since no yield data are available. In the same regard, the comparison of eqn 1 with other allometric equations is not useful, since you never know the true AGB for the plots. If this comparison shall be kept, then please change it into some kind of uncertainty estimate. Rˆ2 values do not help much here, since all equations are based on the same parameter (diameter), so please report RMSE values. Related: in fig. S3, please provide equidistant scaling of the axes.

- L191 ff: To me, it is unclear how to relate the DBH of a single tree to area-based basal area estimate. Please elaborate here.

- L197: You are not using a D-H model.

2) Calculation of structural diversity

- L210ff: Why do you optimise for a good correlation between H for DBH and height? If you so, you might as well use only one of these factors as a surrogate variable for general tree dimension diversity. I suggest comparing results for different discretisation cutoffs instead. This would also interesting for the SEM approach: stand age drives structural diversity, but the direct link between stand age and C-stocks is stronger than the indirect one. One reason for this might be a mismatch in classification resolution.

3) Statistical analysis

- You present a variety of linear modeling variants, when all you want to know is how a set of six parameters influences two response variables. The first set of analysis is contained in the second set, and the second set is a complicated way of doing an AIC-based stepwise procedure (under the assumption that collinearity in the design matrix

is manageable, which you suggest, but might want to reconsider given the explained variance of the single predictors sum up to > 160% (see L330ff)).

- The basic question, as I understand it, is: which set of variables is the best choice for predicting C-stocks. Following this logic, a validation approach would be suited to address the problem, either using a stepwise procedure, using explicit variants of multiple regression models (like already done for the second stream of analyses), or a learning routine that allows for inspection of relative variable importance (like random forests). 80 plots could well be enough for such a validation scheme.

The results are presented in a clear and concise fashion, and the discussion is consistent, comprehensible and linked to current literature, given the results based on the complex analysis scheme.

Some minor corrections:

- L339 "range" instead of "ranged"

- L480 "which was also found"

- L537 "to increase C storage"

- L187 "using Brown's"

- L190 why switch from DBH to D?

- L192 "using Brown's"

- L194 "that Brown's"

- L201 AGBt

- L247 "using equation 3"

---

## Referee Comment (RC2) · Anonymous Referee #2 · 28 Mar 2016

General comments

In general, I consider the MS has great potential in providing a strong contribution to ecological literature by assessing the relative role of different predictors and particularly of structural and species diversity on carbon stocks in subtropical secondary forests. This is a topic of active research today. However, I consider the current version is still away from publishable in Biogeosciences. I have five main comments on this:

1) First, Rather than providing a strong conceptual approach for framing their aim, that is, testing the role of structural diversity on aboveground biomass, authors made a long but not structured literature review of the many variables that could explain variation in AGC stocks, of course making particular emphasis on those the will further

test. After such review, there are no clear stated hypothesis guiding the application of statistical methods and their prediction is so general and non-exclusive that it could be demonstrable almost with any result. I consider the conceptual model in Figure 1 is a good starting point, but such a model should be clearly sustained in the introduction. It could serve as the hypothesis to be tested. Another argument on favor of this critique is that soil carbon stocks are almost no introduced and furthermore, authors pretend to explain them with the same set of predictors than used for the AGC case. This shows a naive approach that does not take into account the vast literature on the factors influencing C stocks in (tropical) soils.

2) In accordance with the unstructured introduction, authors present a wide range of statistical tests for testing basically the same idea. They use simple linear regression, multiple regression and SEM to test the same predictors each time. If you have worked to present a conceptual model like that in Figure 1, why to use approximations do does not allow to test it? Moreover, simple and multiple regressions ended providing almost the same results that SEM, with the exception of two new significant interactions in the SEM model, which are then undervalued by the authors. So I would suggest that according to the idea of a very clearly presented unique hypothesis, a unique analysis should be presented, in which case SEM seems to be the best option.

3) There are some parts of the discussion where authors present possible explanations to their results, but they do not realize that their own results (particularly the SEM) provide no support for such explanations. I consider that a more careful interpretation of such a model should be done.

4) Authors sometimes cite references that are not appropriate or even not refer to the point under discussion. See several specific comments below.

5) I consider the inclusion of site productivity as a predictor should be reconsidered (see specific comments below).

Specific comments

[Figure]

Line 54. Replace ", and store" by "by capturing ". Yu et al 2014 highlight the capture capability rather than the currents C stocks.

Lines 58-59. Authors assert site productivity impact C stocks. However, Lohbeck et al. did't tested the effect of site productivity on biomass or carbon stocks, they tested the reverse. A recent test of the effect of productivity on biomass can be found in Grace et al. 2016 Nature for grasslands, or the general hypothesis for the causal relations between productivity and biomass in tropical forests can be found in Quesada et al. 2012 Biogeosciences or Malhi et al 2012 J. of Ecology

Line 60. Does species diversity impact C stocks? The reference provided (Con et al. 2013) does not seem to provide conclusive evidence. I suggest to soften this assertion and to look for additional literature to sustain it. See for example Cardinale et al. 2011. Am. J. Bot.

Lines 61-63. Although authors use consistently a definition of "stand structural charac-teristics" throughout the MS which includes both "structural" and "diversity" variables, I consider this concept does not provide to the reader a complete idea of what is being tested here, and could hamper the interest on the work. The role of biodiversity has been the subject of much research in the last two decades and stating it separately may make more appealing the work to a broader audience. Therefore, I would suggest to use different concepts for structure and diversity.

Line 66. Include the recent work from Poorter et al. 2016 in Nature "Biomass resilience of secondary forests"

Line 69. I would say that Age is a variable that summarizes or reflects the action of several processes. Probably the authors need to rethink how age is included in their conceptual model. Particullarly, which would be the direct effect of stand age on carbon stocks? What is the ecological mechanisms behind such effect?

Lines 78-82. Soil C is an important component of the study. However, it is just briefly

introduced and the ecological mechanisms linking aboveground biomass or productivity with soil C stocks are not explained here. Therefore, your questions regarding soil C are not fully understandable.

Lines 83-90. These lines say the same than previous paragraphs, no? Probably better to merge them with previous paragraphs and to try to focus more on the general hypothesis regarding the effects of forest age, stand structure and stand diversity.

Line 98. What is C synthesis?

Lines 110-111. So anything could explain C stocks? Isn't there a hypothesis on which of this potential explanatory variables could be more important? Also, what is stand density? Isn't it included within stand structure in general?

Line 112. What is a direct effect of stand age? Isn't it mediated always by stand charachteristics? Which is its ecological basis?

Lines 114-115. This generalization applies only for wet forest, probably not for dry forests. Please be specific.

Lines 117-118. That is not an adequate prediction, that is a "all matters" scenario. Rather, say that you tested the contribution of different predictors.

Line 122. Randomly? Within the entire landscape? How were you sure they represented all the successional gradient possible? There were no mature forests, conserved and/or degraded? Did you use a GIS to select them? Please elaborate on site selection.

Line 122. Stand age in relation to what? What kind of disturbance?

Lines 124-130. Questions should be rephrased, their actual form is not appealing (they seem barely descriptive). Also, questions 1 and 2 are the same but in their discrete and continuous forms, respectively.

Line 140. The "consequently" is not clear. Authors asserted "there were different

intensities of human disturbances (typically logging)" Do they refer to different types of disturbance, different intensities of logging, or both? This is quite important since recent studies on succession have highlighted the relevance of different types of previous land-use or land-use intensities for the unfold of succession (Mesquita et al. BioSciences 2015, Arroyo-Rodríguez 2015 Biological Reviews). Moreover, it is particullarlly relevant the authors provide a detailed description of the disturbance history of the region and of the related criteria for selecting plots in particular.

Line 141. Rather than developmental stages, which may refer to a departure from a clear-cutted forests, authors could use "stands with different levels of degradation" or "stands with different level of perturbation"

Line 142. Does this mean that there was previously a landscape characterization of different landcover types from which it was possible to filter only successional forests and to select randomly the location of the plots?

Line 143. Any kind of disturbance? Excluding only recent human disturbance? What do authors mean exactly by "recent"?

Line 148. What do the authors mean by "typical habitats"? Did the authors include plots in different environmental conditions? Or do they refer to different successional habitats all in under the same environmental conditions?

Line 152. It is interesting that until here I assumed the authors constructed a chronosequence of sites derived from a pulse-type disturbance. This was probably because of the use of the terms forest age and secondary forests, which are commonly used in the literature to refer to clear-cutted sites. However, after looking at Table 1, I figured out that sites were assigned to one of three different "development stages", which seems to be different in the intensity of previous logging. Therefore, sites were not clear-cutted but instead affected by a pressure-disturbance like continuous logging. Therefore, I suggest the authors provide their working definition of secondary forest, or, alternatively, use the term "degradation level", "degradation intensity" or simply "logging intensity" to refer to their different levels of logging. Authors can look at several references for the definitions of secondary forest and degraded forest (Chazdon 2014 Second Growth, Chapter 1; Chokkalingam & de Jong 2001 International Forestry Review; Putz & Redford 2010 Biotropica).

Line 169. Which stages? You have not defined such stages here.

Lines 170-171. Ok, so it is an indirect measure of productivity. Much more is therefore required on the definition of the disturbance regime to which such plots were subjected. Was the initial point (year 0) a clear-cutted forest for all? Or a selectively logged forest as suggested by Table 1?

Line 176. Which one of these references was used to calculate biomass? Please be specific.

Lines 175-184. Why is this paragraph here? A portion could be used during model framing in the introduction section.

Lines 188-189. This is not an argument to exclude height from biomass calculation. See for example Chave et al. 2014 GCB for a detailed discussion on height inclusion in allometric equations.

Line 192. First sentence is not clear: what kind of uncertainty is avoided and why?

Line 193. second sentence should be re-written

Line 196-197. what are D-H models?

Lines 210-211. Why you did not use the Chave et al. 2014 equation, which seems to improve Chave's et al 2005 equations?

Line 2015. Therefore, which equation you used? I suggest all this discussion could go in an Annex or supplementary material, leaving here in the methods only the descripcion of the equation finnally used

Line 216. Why you did not used the Alí et al. 2015 equation for all the tree community?

Line 239. Does this values refers to the number of categories, the range of the categories or the limits of the categories?

Lines 244-245. Why to use correlated DBH-height classes if you then want to assess their explanatory ability in a unique multiple regression model? Should not the categories be selected based on their correlation to the variables you want to explain, i.e. biomass? You could simply try to test correlation between diversity and biomass and select those categorizations given the maximum correlation.

Line 251. Mathematical notation is wrong. x should denote only one thing: or the number of different attributes evaluated (3) or the number of classes within a attribute. Furthermore, sub-index for p should be i (pi), because the proportion is evaluated for each i class within 1 and x (if x is the total number of classes).

Line 270. Please say explicitly at the beggining of the secition 2.3 which C pools are considered in this study: "two carbon pools were assessed in this study: aboveground living biomass of the tree community (excluding lianas and herbs, no'), and soil organic C in the top 20 cm of soil").

Lines 270-276. Probably better to summarize lines 270-276 by saying that for each series, al the possible variable combinations and interactions were tested (a fully ...model) and the best model was selected using AIC.

Line 291. If you have previously settled a hypothesis of a hierarchy of effects acting on C stocks, why to use simple and multiple linear models and not going directly to the SEM? What is the original hypothesis? Doesn't SEM allows you to test the same that multiple regression model allows, that is, which are the structural determinants of the C stocks?

Line 304. Age is not expected to be linearly related to AGC. Also, from Figure 2 it seems that some of the relations could be better explained using a non-linear (but

probabliy linearizable) model.

Lines 307-310. So, really the logic behind fitting such models was to select the best to use in SEM? Why not allowing SEM to test the whole model? Why testing two different models if you can test only one?

Lines 314-320. This paragraph is very difficult to grasp. Does the second sentence mean that rather the structural diversity, the proportion of big trees could alternatively explain biomass?

Lines 315-318. If I understood well, this is the same problem with analyzing Shannon index results for species diversity: we do not actually know if an increased diversity is caused by increased number of categories (which in this case means increased number of big trees) or by a more even distribution among categories (that is, basal area is more equitatively distributed among dbh categories). If you want to dissect such effects, then wouldn't be easier to have from the beginning to different predictors indicating directly such different possible explanations? Moreover, previous findings would allow authors to hypothesize that the amount of big trees is an important predictor of forest biomass (Slik et al. 2013 Global Ecol. Biogeo.), so authors could use some indicator of the size of the biggest trees as a predictor of biomass.

Line 322. I'm not completely sure that a higher correlation with CV means that dominance of big trees is not important. Higher CV values means that deviation from the mean DBH or H increases, which can happen if bigger trees are present but there is an uneven size distribution.

Line 332. Most of the significant relations seems to violate linear regression assumptions, particularlly that the straight line is an adequate representation of the relationchip or that variance is homogeneous. Authors do not clarify through the text or in the supplemetary tables if other relationships were tested or if variables were transformed to meet assumptions

Line 334. Species density? Stand density?

Line 341. What is the positive variation?

Lines 360-363. Probably, the synthetic models are not necesary. Authors can check that the relative importance of variables in the synthetic modelcorrelates negatively but perfectly to the p values associated to each of the variables in the best-fit model. So probably that part could be taken to the supplementary material.

Lines 368-369. As expected, there is no direct functional relation between stand characteristics and C stocks. This only reflects the poor literature review on the mechanisms that drive C accumulation in tropical forests soils.

Lines 377-379. There is no sense in having these two alternative models, at least if there are no competing hypothesis grounded on strong ecological knowledge.

Lines 380-381. I really have a doubt on the meaning of the variable "productivity" here. As defined, productivity is calculated on the basis of stand volume divided by forest age. Stand volume is another measure of biomass (the volume of a forest is filled with biomass, so as it is bigger, biomass is bigger), rather than an "independent" structural measurement. I really think that it is an spurious relation and that the authors should consider to exclude it from the model.

Lines 382-383. What is the difference between this model and the multiple regression model?

Lines 410-411. This last sentence evidence the poor literature review made by the authors on the ecological and physical processes controlling C stocks in soils. I suggest to not include soil C stock estimation in the model, but rather to provide their estimates as supplementary material.

Lines 419-420. Such argument would imply that higher species diversity have incidence on higher structural diversity. However, there is no association between species and DBH diversity, so data does not support such possibility.

Line 433. If such argument was true, a significant relation between species diversity and stand age should arise.

Line 449. Uncertain? It seems authors are "averaging" results from two different approaches and therefore saying that there is no conclusive evidence, even with the same data! That's why it is important to have a clearly stated hypothesis from the beginning and to use the adequate analytical framework to test it.

Line 451. A similar argument was raised by Grace et al. 2016 Nature

Lines 467-468. Site productivity does not mediate such relation according to SEM. Please rephrase.

Line 481. Dupuy et al. 2012 do not test age as a predictor of biomass. Please see Hernández-Stefanoni et al. 2010 Landscape Ecology for the adequate reference. There are a lot more of references on the recovery of biomass or AGC stock during succession in both wet and dry tropical forests. See also Poorter et al. 2016 Nature for a recent compendium.

Lines 485-487. this argument is not right. Although it is true that at tree level bigger trees acumulate more carbon, at the stand level it is not true if we have a gradient of forest age, for which maximum accumulation commonly occurs early in succession. See Mora et al. 2016 Biotropica, Vargas et al 2008 GCB or Yang et al. 2011 New Phytologist for how expected rates of change should be higher in the first decades of succession.

Line 488. Not pretty sure of this since CV test does not seem to be the best indicator.

Line 499. Lohbeck et al. 2015 never tested productivity as a predictor of biomass, but the reverse (biomass as a predictor of productivity).

Lines 500-504. In the model site productivity is not affected by forest age, so this argument does not march data.

Line 514-516. This argument is not clear at all

Lines 536-537. Please elaborate more on how stand diversity could be improved based on your results.

Line 790. Why should soil organic C depend on structural stand variables? There are many ecological process between C accumulation in the aboveground biomass and its accumulation in soil (literfall, biomass decay, microbial growth), plus a set of factors that may have greater potential impact (soil type, bulk density, previous land use, etc). For the case of soil organic C, this model seems very naive.

Specific comments

Line 123. Replace "in accordance to" by "regarding the" or "about the"

Line 230. Delete "in"

Line 247. Please modify to ".. diversities were calculated for each plot using equation 3".

Line 254. Replace "analysis" by "calculation"

Line 512. Replace by "effect"

---

## Author Comment (AC1) · 9 May 2016

Ali et al. present a study on an interesting and important topic: biomass estimation for subtropical forests in the East Asian monsoon region. The study is generally well intraoduced and clearly structured. The data set is most probably appropriate to tackle the research questions raised by the authors. The choice of analytical methods, however, needs considerable reconsideration in some regards.

=> We are grateful to referee #1 for providing useful comments on our study. We will follow your suggestions and those recommended by referee #2 to revise this manuscript (MS). According your constructive comments, we have reorganized the conceptual models (please see attachment) by considering comments from referee #2 as well.

In addition, we re-analyzed our data with SEM model and we believe that this MS will be improved thoroughly. Thank you.

Please find our responses to your specific comments below.

1) Measurements and calculations of carbon stocks - There are no measurements of carbon stocks, just calculations based on allometric equations, so please adjust the section title accordingly.

=> We will adjust the section title. Thank you.

- I was not able to find eqn 1 in Brown et al. 1989, please indicate exact reference or modification if applicable.

=> Actually, we used the revised form of Brown et al.'s (1989) equation, which had been published in FAO papers (1997). We apologize for wrong citation. In the new revision, we will estimate AGB by using Chave et al. (2014) equation in the revise MS, based on the comments of referee #2.

- 14% of variance in tree height are not explained by diameter. This information could be used to improve allometric estimates, since the diameter-height-allometry varies with environmental conditions, and might provide valuable additional information.

=> This is a constructive comments. In the new revision, we will employ Chave et al. (2014) model by using DBH, H and wood density as predictors, and we believe that this model can improve the estimation of AGB of big trees. Actually, we have finished the data analysis by following Chave et al. (2014) model.

- However, there is no way of validating your AGB estimates, since no yield data are available. In the same regard, the comparison of eqn 1 with other allometric equations is not useful, since you never know the true AGB for the plots. If this comparison shall be kept, then please change it into some kind of uncertainty estimate. RËĘ2 values do not help much here, since all equations are based on the same parameter (diameter), so please report RMSE values. Related: in fig. S3, please provide equidistant scaling

of the axes.

=> We agree with your comment that we cannot validate AGB estimates in the previous MS. We will use Chave et al. (2014) equation, as it has been found to be the most suitable and appropriate equation for tropical and subtropical forests. Therefore, there will be no need to compare AGB estimates from different allometric equations. However, we will validate our AGB estimates by conducting correlation with stand basal area. Thank you.

- L191 ff: To me, it is unclear how to relate the DBH of a single tree to area-based basal area estimate. Please elaborate here.

=> Sorry for lack of clarity in the previous version of our manuscript. Tree basal area is pi*(DBH/2)^2, and stand basal area is the sum of all tree basal area. We will provide a correlation figure between stand basal area and AGB per plot, and fitted the regression line with Type II - RMA. Thank you.

- L197: You are not using a D-H model.

=> We will clarify this in the revised MS.

2) Calculation of structural diversity - L210ff: Why do you optimise for a good correlation between H for DBH and height? If you so, you might as well use only one of these factors as a surrogate variable for general tree dimension diversity. I suggest comparing results for different discretization cutoffs instead. This would also interesting for the SEM approach: stand age drives structural diversity, but the direct link between stand age and C-stocks is stronger than the indirect one. One reason for this might be a mismatch in classification resolution.

=> We agreed with your suggestion, we will compare results for different discretization by employing SEM models and select the best model through AIC. For example, we will use different combinations of height and DBH diversities based on different discrete classes in the SEM models and then select the best model through AIC. Moreover, in

the new revision, we will use structural diversity as a latent variable that includes both DBH and height diversity indices. Before making this reply, we indeed have done data re-analysis by following the approach mentioned above.

3) Statistical analysis - You present a variety of linear modeling variants, when all you want to know is how a set of six parameters influences two response variables. The first set of analysis is contained in the second set, and the second set is a complicated way of doing an AIC based stepwise procedure (under the assumption that collinearity in the design matrix is manageable, which you suggest, but might want to reconsider given the explained variance of the single predictors sum up to > 160% (see L330ff)). - The basic question, as I understand it, is: which set of variables is the best choice for predicting C-stocks. Following this logic, a validation approach would be suited to address the problem, either using a stepwise procedure, using explicit variants of multiple regression models (like already done for the second stream of analyses), or a learning routine that allows for inspection of relative variable importance (like random forests). 80 plots could well be enough for such a validation scheme.

=> Thanks for your constructive comments here. We would like to follow your comments on diversities and compare the results. Therefore, we will only use SEM model for comparing different models based on different combinations of DBH and height diversities of different discrete classes. Further, we will also redesign our conceptual model in order to test the complex paths in one SEM model, instead of two models (as conducted in the last MS).

The results are presented in a clear and concise fashion, and the discussion is consistent, comprehensible and linked to current literature, given the results based on the complex analysis scheme.

=> Thanks a lot!

Some minor corrections: - L339 "range" instead of "ranged" - L480 "which was also found" - L537 "to increase C storage" - L187 "using Brown's" - L190 why switch from

DBH to D? - L192 "using Brown's" - L194 "that Brown's" - L201 AGBt - L247 "using equation 3"

=> We will correct all these mistakes in the revised MS. Thank you.

—————————————————————

a

[Figure]

b

[Figure]

c

[Figure]

**Fig. 1.**

---

## Author Comment (AC2) · 9 May 2016

General comments In general, I consider the MS has great potential in providing a strong contribution to ecological literature by assessing the relative role of different predictors and particularly of structural and species diversity on carbon stocks in subtropical secondary forests. This is a topic of active research today. However, I consider the current version is still away from publishable in Biogeosciences. I have five main comments on this:

=> We are grateful to referee #2 for providing constructive comments on our manuscript. According to your comments, we will thoroughly revise the MS both in theoretical and analytical aspects. Please finding our responses to your specific comments below.

1) First, Rather than providing a strong conceptual approach for framing their aim, that is, testing the role of structural diversity on aboveground biomass, authors made a long but not structured literature review of the many variables that could explain variation in AGC stocks, of course making particular emphasis on those the will further test. After such review, there are no clear stated hypothesis guiding the application of statistical methods and their prediction is so general and non-exclusive that it could be demonstrable almost with any result. I consider the conceptual model in Figure 1 is a good starting point, but such a model should be clearly sustained in the introduction. It could serve as the hypothesis to be tested. Another argument on favor of this critique is that soil carbon stocks are almost no introduced and furthermore, authors pretend to explain them with the same set of predictors than used for the AGC case. This shows a naive approach that does not take into account the vast literature on the factors influencing C stocks in (tropical) soils.

=> Thanks for this constructive comments. We agree with your concerns that the research aims are not well structured in our previous manuscript. In the new revision, we will clearly introduce our conceptual model in the introduction for driving the specific hypothesis. We will test new conceptual models (please attached figure) in the revised manuscript, by considering your suggested papers in the following comments. The best SEM model among all combinations of DBH and height diversity based on different discrete classes and/or three types of paths between species diversity and structural diversity will be selected on the basis of lowest AIC. We will exclude stand density (trees per hectare) and site productivity from our conceptual model, as you have suggested in the following comments. => In addition, we have re-analyzed our data with SEM models before we make this reply and we believe that revise MS will be improved thoroughly by testing the models below.

2) In accordance with the unstructured introduction, authors present a wide range of statistical tests for testing basically the same idea. They use simple linear regression,

multiple regression and SEM to test the same predictors each time. If you have worked to present a conceptual model like that in Figure 1, why to use approximations do does not allow to test it? Moreover, simple and multiple regressions ended providing almost the same results that SEM, with the exception of two new significant interactions in the SEM model, which are then undervalued by the authors. So I would suggest that according to the idea of a very clearly presented unique hypothesis, a unique analysis should be presented, in which case SEM seems to be the best option.

=> We agreed with your comments about statistical analysis. We will use SEM models by testing different combinations of height and DBH diversities based on different discrete classes, and then select the best model through AIC. In this way, we believe that our proposed hypothesis and conceptual model will be clear than the last version of the MS. Actually, we have finished the data analysis by following the approach above when we write this reply. We found that this approach is feasible and reasonable for testing the specific hypothesis.

3) There are some parts of the discussion where authors present possible explanations to their results, but they do not realize that their own results (particularly the SEM) provide no support for such explanations. I consider that a more careful interpretation of such a model should be done.

=> We apologize for the lack of clarity in the discussion section. We will clearly discuss our model with sound evidences in this study and other studies. Thank you.

4) Authors sometimes cite references that are not appropriate or even not refer to the point under discussion. See several specific comments below.

=> We will avoid such mistakes in the revised MS. We apologize for inappropriate citations in the discussion section.

5) I consider the inclusion of site productivity as a predictor should be reconsidered (see specific comments below).

=> Thanks for your constructive comment. We will follow your suggested paper (Grace et al. 2016 Nature for grasslands) for making a new conceptual model, as already shown above. Therefore, site productivity will be excluded as a predictor.

Line 54. Replace ", and store" by "by capturing ". Yu et al 2014 highlight the capture capability rather than the currents C stocks.

=> We will correct in the revised MS. Thank you.

Lines 58-59. Authors assert site productivity impact C stocks. However, Lohbeck et al. did't tested the effect of site productivity on biomass or carbon stocks, they tested the reverse. A recent test of the effect of productivity on biomass can be found in Grace et al. 2016 Nature for grasslands, or the general hypothesis for the causal relations between productivity and biomass in tropical forests can be found in Quesada et al. 2012 Biogeosciences or Malhi et al 2012 J. of Ecology

=> Thanks for pointing it out. Actually, this is a wrong citation. We will revise this and make new concept model by considering your suggested papers in the revised MS. Thank you.

Line 60. Does species diversity impact C stocks? The reference provided (Con et al. 2013) does not seem to provide conclusive evidence. I suggest to soften this assertion and to look for additional literature to sustain it. See for example Cardinale et al. 2011. Am. J. Bot.

=> Thanks for your constructive comment here, and we will soften this assertion and revised it by following your suggestion accordingly.

Lines 61-63. Although authors use consistently a definition of "stand structural characteristics" throughout the MS which includes both "structural" and "diversity" variables, I consider this concept does not provide to the reader a complete idea of what is being tested here, and could hamper the interest on the work. The role of biodiversity has been the subject of much research in the last two decades and stating it separately

may make more appealing the work to a broader audience. Therefore, I would suggest to use different concepts for structure and diversity.

=> This comment is very constructive. We will follow your suggestion by considering stand structural diversity as a latent variable including DBH and height diversity, while species diversity as a separate variable, as shown in the above mentioned conceptual models. Thank you.

Line 66. Include the recent work from Poorter et al. 2016 in Nature "Biomass resilience of secondary forests"

=> We will consider their recent work, and thanks for suggesting this significant paper.

Line 69. I would say that Age is a variable that summarizes or reflects the action of several processes. Probably the authors need to rethink how age is included in their conceptual model. Particullarly, which would be the direct effect of stand age on carbon stocks? What is the ecological mechanisms behind such effect?

=> Indeed, Age is a variable that summarizes processes such as growth, ingrowth and mortality. Our data do not include processes-based measurements, and we wish to use age to summarize multiple processes responsible for standing above-ground carbon. We will use one type of complex conceptual model in the revised MS. In the last version of MS, it was quite a big confusion by using two different models, such as age model and stand characteristics model. We will avoid such type of confusion in the revised model. Thank you.

Lines 78-82. Soil C is an important component of the study. However, it is just briefly introduced and the ecological mechanisms linking aboveground biomass or productivity with soil C stocks are not explained here. Therefore, your questions regarding soil C are not fully understandable.

=> After careful consideration, we feel that it may be best to exclude SOC component since data associated with many drivers such as local site condition, past disturbance

history as well as litterfall (leaves and roots) feedback for SOC are not available. We would like to use much of our efforts on AGC stock by testing numbers of SEM models, in order to clarify the effects of stand age, stand structural diversity and species diversity on AGC stock.

Lines 83-90. These lines say the same than previous paragraphs, no? Probably better to merge them with previous paragraphs and to try to focus more on the general hypothesis regarding the effects of forest age, stand structure and stand diversity.

=> We will introduce our new hypothesis based on our new proposed conceptual models, as you have suggested in the last comments. Thank you.

Line 98. What is C synthesis?

=> We apologize for using different terminology here. Actually, we meant C stock or storage.

Lines 110-111. So anything could explain C stocks? Isn0t there a hypothesis on which of this potential explanatory variables could be more important? Also, what is stand density? Isn't it included within stand structure in general?

=> Thanks for your constructive comment. In the revised MS, we will consider stand structural diversity including DBH and height diversity, and species diversity as potential explanatory variables, when assessing the residual effect of stand age on both of them. Stand density is the number of trees per hectare. Yes, it is included within stand structure in general. We will avoid this variable in the revised conceptual models.

Line 112. What is a direct effect of stand age? Isn0t it mediated always by stand charachteristics? Which is its ecological basis?

=> With increased stand age, biomass accumulation will increase by following stand development, tree growth and increased stand structural diversity. Therefore, stand age can act as a driver for improving carbon stocks. In the revised MS, we will use one complex conceptual model. In the last version of MS, it was quite a big confusion by

using two different models, such as age model and stand characteristics model. We will avoid such type of confusion in the revised model. Thank you.

Lines 114-115. This generalization applies only for wet forest, probably not for dry forests. Please be specific.

=> Yes, we will correct here.

Lines 117-118. That is not an adequate prediction, that is a "all matters" scenario. Rather, say that you tested the contribution of different predictors.

=> We will design a new conceptual model by considering all your comments on the previous conceptual models. Thank you.

Line 122. Randomly? Within the entire landscape? How were you sure they represented all the successional gradient possible? There were no mature forests, conserved and/or degraded? Did you use a GIS to select them? Please elaborate on site selection.

=> Thanks for pointing it out. 'Randomly' is not an appropriate description of site selection. Actually, we selected site and plot through both field survey and local forestry inventory that used for classifying regional vegetation types. We will further elaborate on site selection in the revised MS.

Line 122. Stand age in relation to what? What kind of disturbance?

=> We define stand age as time since last stand replacing disturbance, which includes clearcutting, reclamation from agriculture, and typhoon. This will be clarified.

Lines 124-130. Questions should be rephrased, their actual form is not appealing (they seem barely descriptive). Also, questions 1 and 2 are the same but in their discrete and continuous forms, respectively.

=> We will revise the proposed questions according to the new hypothesis and conceptual model. Thank you.
Line 140. The "consequently" is not clear. Authors asserted "there were different intensities of human disturbances (typically logging)" Do they refer to different types of disturbance, different intensities of logging, or both? This is quite important since recent studies on succession have highlighted the relevance of different types of previous land-use or land-use intensities for the unfold of succession (Mesquita et al. BioSciences 2015, Arroyo-Rodríguez 2015 Biological Reviews). Moreover, it is particullarlly relevant the authors provide a detailed description of the disturbance history of the region and of the related criteria for selecting plots in particular.

=> Yes, different types of disturbances such as logging, land conversion, typhoon etc, as well as different intensities of logging at different sites were happened in the history. We will clarify here in the revised MS. We will provide a detailed description of the disturbance history and criteria for plot selection. Thank you.

Line 141. Rather than developmental stages, which may refer to a departure from a clear-cutted forests, authors could use "stands with different levels of degradation" or "stands with different level of perturbation"

=> We will clarify here in the revised MS. Thank you.

Line 142. Does this mean that there was previously a landscape characterization of different landcover types from which it was possible to filter only successional forests and to select randomly the location of the plots?

=> Yes, more exactly saying, there was a landscape characterization of different forest use types, i.e., secondary shrublands, mature forests protected from clearcutting or logging, and logging forest. We will clarify this section in the revised MS. Actually, the detail description of the study area was not included in the last version of MS, which make reviewers doubtful.

Line 143. Any kind of disturbance? Excluding only recent human disturbance? What do authors mean exactly by "recent"?

=> We will clarify the kind of disturbance in the new revision. Recent means for the last 3 decades according records from local government. Thank you.

Line 148. What do the authors mean by "typical habitats"? Did the authors include plots in different environmental conditions? Or do they refer to different successional habitats all in under the same environmental conditions?

=> Sorry for the vague wording, we will rephrased this statement. Thank you.

Line 152. It is interesting that until here I assumed the authors constructed a chronosequence of sites derived from a pulse-type disturbance. This was probably because of the use of the terms forest age and secondary forests, which are commonly used in the literature to refer to clear-cutted sites. However, after looking at Table 1, I figured out that sites were assigned to one of three different "development stages", which seems to be different in the intensity of previous logging. Therefore, sites were not clear-cutted but instead affected by a pressure-disturbance like continuous logging. Therefore, I suggest the authors provide their working definition of secondary forest, or, alternatively, use the term "degradation level", "degradation intensity" or simply "logging intensity" to refer to their different levels of logging. Authors can look at several references for the definitions of secondary forest and degraded forest (Chazdon 2014 Second Growth, Chapter 1; Chokkalingam & de Jong 2001 International Forestry Review; Putz & Redford 2010 Biotropica).

=> Thanks for your constructive and helpful comments on site selection. We will clarify here in the revised MS, by following your comments and suggested papers for definitions.

Line 169. Which stages? You have not defined such stages here.

=> Developmental stages such as young, pre-mature and mature forests. We will define here in the revise MS.

Lines 170-171. Ok, so it is an indirect measure of productivity. Much more is therefore

required on the definition of the disturbance regime to which such plots were subjected. Was the initial point (year 0) a clear-cutted forest for all? Or a selectively logged forest as suggested by Table 1?

=> Yes, we have indirectly estimated the site productivity by reviewing the official documents of Ningbo Forestry Bureau, Zhejiang Province, to collect relevant data about the disturbances for each site in the study area. The study plots included both clear-cutted forests and selectively logged forests. More exactly saying, there was a landscape characterization of different forest use types, i.e., secondary shrublands, mature forests protected from clearcutting or logging, and logging forest.

Line 176. Which one of these references was used to calculate biomass? Please be specific.

=> We used both references because Brown's (1989) equation only covers trees with DBH > 5 cm while Ali et al. (2015) equation was developed for small trees and shrubs. Thank you.

Lines 175-184. Why is this paragraph here? A portion could be used during model framing in the introduction section.

=> Thanks for constructive suggestion here. We will revise MS as recommended.

Lines 188-189. This is not an argument to exclude height from biomass calculation. See for example Chave et al. 2014 GCB for a detailed discussion on height inclusion in allometric equations.

=> We will use Chave et al. (2014) GCB equation for the estimation of AGB in the revised MS. Thank you.

Line 192. First sentence is not clear: what kind of uncertainty is avoided and why? Line 193. second sentence should be re-written

=> These sentences will be rewritten. Actually, most of the generalized allometric

equations are for tropical forests instead of subtropical forests. Therefore, we compared different models to avoid uncertainty. We will used Chave et al. 2014 equation, which is extendable to subtropical regions. Thank you.

Line 196-197. what are D-H models?

=> Model using DBH and height as predictors for estimation of AGB. We will clarify in the revised MS.

Lines 210-211. Why you did not use the Chave et al. 2014 equation, which seems to improve Chave0s et al 2005 equations?

=> We will use Chave et al. 2014 equation in the revised MS. Thank you.

Line 2015. Therefore, which equation you used? I suggest all this discussion could go in an Annex or supplementary material, leaving here in the methods only the description of the equation finnally used

=> Brown's equation was used for the estimation of AGB of big trees. Now, we will use Chave et al. 2014 equation in the revised MS. Thank you.

Line 216. Why you did not used the Alí et al. 2015 equation for all the tree community?

=> Ali et al. 2015 equations were only developed for small trees and shrubs.

Line 239. Does this values refers to the number of categories, the range of the categories or the limits of the categories?

=> These values refers to the limits of the categories. For example, for DBH 0 – 2 cm, 2.1 – 4 cm etc.

Lines 244-245. Why to use correlated DBH-height classes if you then want to assess their explanatory ability in a unique multiple regression model? Should not the categories be selected based on their correlation to the variables you want to explain, i.e. biomass? You could simply try to test correlation between diversity and biomass and

select those categorizations given the maximum correlation.

=> Thanks for your constructive comment. By following this comment, we can not get any good fit for SEM model, as we have tried. Therefore, it is better to test different SEM models instead of just focusing on correlations. In the new revision, we will test a number of SEM models through combinations of different DBH and height diversities based on different discrete classes, and then select the best model through AIC. In order to make things clear, we will provide statistics of all SEM models in one table and more details for selected best model. Thanks.

Line 251. Mathematical notation is wrong. x should denote only one thing: or the number of different attributes evaluated (3) or the number of classes within a attribute. Furthermore, sub-index for p should be i (pi), because the proportion is evaluated for each i class within 1 and x (if x is the total number of classes).

=> We will correct the equation form in the revised MS. Thank you for pointing it out.

Line 270. Please say explicitly at the beggining of the section 2.3 which C pools are considered in this study: "two carbon pools were assessed in this study: aboveground living biomass of the tree community (excluding lianas and herbs, no'), and soil organic C in the top 20 cm of soil").

=> We will include the beginning statement for C stocks.

Lines 270-276. Probably better to summarize lines 270-276 by saying that for each series, al the possible variable combinations and interactions were tested (a fully ...model) and the best model was selected using AIC. Line 291. If you have previously settled a hypothesis of a hierarchy of effects acting onC stocks, why to use simple and multiple linear models and not going directly to the SEM? What is the original hypothesis? Doesn0t SEM allows you to test the same that multiple regression model allows, that is, which are the structural determinants of the C stocks?

=> By considering your all comments on the conceptual model, we will only employy

SEM model in the revised MS. In addition, the bivariate relationships will be included in the appendix file or even main text. Thank you.

Line 304. Age is not expected to be linearly related to AGC. Also, from Figure 2 it seems that some of the relations could be better explained using a non-linear (but probabliy linearizable) model.

=> We will consider your suggestion in the revised MS, by assessing both linear and several linearizable forms (log, exponential and 2nd order polynomial) and choose the one with lowest AIC in our revised SEM. Thank you.

Lines 307-310. So, really the logic behind fitting such models was to select the best to use in SEM? Why not allowing SEM to test the whole model? Why testing two different models if you can test only one?

=> We will use one SEM model and access the whole model as well as the best model based on AIC. Thanks for helpful suggestion.

Lines 314-320. This paragraph is very difficult to grasp. Does the second sentence mean that rather the structural diversity, the proportion of big trees could alternatively explain biomass?

=> Yes, you are right. We will correct here in the revised MS.

Lines 315-318. If I understood well, this is the same problem with analyzing Shannon index results for species diversity: we do not actually know if an increased diversity is caused by increased number of categories (which in this case means increased number of big trees) or by a more even distribution among categories (that is, basal area is more equitatively distributed among dbh categories). If you want to dissect such effects, then wouldn0t be easier to have from the beginning to different predictors indicating directly such different possible explanations? Moreover, previous findings would allow authors to hypothesize that the amount of big trees is an important predictor of forest biomass (Slik et al. 2013 Global Ecol. Biogeo.), so authors could use some

indicator of the size of the biggest trees as a predictor of biomass.

=> Thanks for your constructive comments here. We will use SEM model to test different combination of DBH and height diversities based on different discrete classes, to know whether increased diversity caused by increased number of categories has any different effect on C stock.

Line 322. I0m not completely sure that a higher correlation with CV means that dominance of big trees is not important. Higher CV values means that deviation from the mean DBH or H increases, which can happen if bigger trees are present but there is an uneven size distribution.

=> We will use the alternative approach, as you have suggested above. Thank you.

Line 332. Most of the significant relations seems to violate linear regression assumptions, particularlly that the straight line is an adequate representation of the relationchip or that variance is homogeneous. Authors do not clarify through the text or in the supplementary tables if other relationships were tested or if variables were transformed to meet assumptions

=> We will provide such results or explanations in the revised MS. Thank you. Line 334. Species density? Stand density?

=> Actually, it was species diversity and stand density (trees per hectare). We will clarify this.

Line 341. What is the positive variation?

=> Means positive linear relationship. We will modify this.

Lines 360-363. Probably, the synthetic models are not necesary. Authors can check that the relative importance of variables in the synthetic model correlates negatively but perfectly to the p values associated to each of the variables in the best-fit model. So probably that part could be taken to the supplementary material.

=> Thanks for constructive comment here.

Lines 368-369. As expected, there is no direct functional relation between stand characteristics and C stocks. This only reflects the poor literature review on the mechanisms that drive C accumulation in tropical forests soils.

=> We will include more potential literatures about AGC stock, while we would like to drop SOC component in the revised MS. Thank you.

Lines 377-379. There is no sense in having these two alternative models, at least if there are no competing hypothesis grounded on strong ecological knowledge.

=> We will redesign a new conceptual model by considering your all comments. Therefore, this part will be updated.

Lines 380-381. I really have a doubt on the meaning of the variable "productivity" here. As defined, productivity is calculated on the basis of stand volume divided by forest age. Stand volume is another measure of biomass (the volume of a forest is filled with biomass, so as it is bigger, biomass is bigger), rather than an "independent" structural measurement. I really think that it is an spurious relation and that the authors should consider to exclude it from the model.

=> We agree with your suggestion to exclude productivity from our conceptual model. Thank you.

Lines 382-383. What is the difference between this model and the multiple regression model?

=> Sorry for providing double proof of the results. We will only consider SEM model in the revised MS.

Lines 410-411. This last sentence evidence the poor literature review made by the authors on the ecological and physical processes controlling C stocks in soils. I suggest to not include soil C stock estimation in the model, but rather to provide their estimates

as supplementary material.

=> OK, we will follow your comments in the revised MS. Thank you.

Lines 419-420. Such argument would imply that higher species diversity have incidence on higher structural diversity. However, there is no association between species and DBH diversity, so data does not support such possibility.

=> We will revise here according to the new analysis. Thank you.

Line 433. If such argument was true, a significant relation between species diversity and stand age should arise.

=> We will revise here accordingly. We apologize for doubtful statement here.

Line 449. Uncertain? It seems authors are "averaging" results from two different approaches and therefore saying that there is no conclusive evidence, even with the same data! That0s why it is important to have a clearly stated hypothesis from the beginning and to use the adequate analytical framework to test it.

=> We apologize for such doubtful argument, it will be clarified in the revised MS. We acknowledge the uncertainty due to using of three types of statistical models. We will focus on SEM model in the revised MS, as you have suggested in above comments.

Line 451. A similar argument was raised by Grace et al. 2016 Nature

=> We will include their argument to support our statement.

Lines 467-468. Site productivity does not mediate such relation according to SEM. Please rephrase. Line 481. Dupuy et al. 2012 do not test age as a predictor of biomass. Please see Hernández-Stefanoni et al. 2010 Landscape Ecology for the adequate reference. There are a lot more of references on the recovery of biomass or AGC stock during succession in both wet and dry tropical forests. See also Poorter et al. 2016 Nature for a recent compendium.

=> We will revise here according to our new conceptual model. Thank you.

Lines 485-487. this argument is not right. Although it is true that at tree level bigger trees acumulate more carbon, at the stand level it is not true if we have a gradient of forest age, for which maximum accumulation commonly occurs early in succession. See Mora et al. 2016 Biotropica, Vargas et al 2008 GCB or Yang et al. 2011 New Phytologist for how expected rates of change should be higher in the first decades of succession.

=> We will correct here in the revised MS. But see Stephenson et al., 2014 (Nature) for such augments, which is really interesting.

Line 488. Not pretty sure of this since CV test does not seem to be the best indicator.

=> For CV of DBH as a good predictor of AGB, please see Zhang & Chen 2015 (J Ecol.). Thank you.

Line 499. Lohbeck et al. 2015 never tested productivity as a predictor of biomass, but the reverse (biomass as a predictor of productivity).

=> We will revise it. Thank you.

Lines 500-504. In the model site productivity is not affected by forest age, so this argument does not march data.

=> We will exclude productivity from our conceptual model, as you have suggested in the one of the above comments. Thank you.

Line 514-516. This argument is not clear at all

=> We will clarify here in the revised MS.

Lines 536-537. Please elaborate more on how stand diversity could be improved based on your results.

=> We will put more light on the importance of stand diversity in the revised MS. Thank

you.

Line 790. Why should soil organic C depend on structural stand variables? There are many ecological process between C accumulation in the aboveground biomass and its accumulation in soil (literfall, biomass decay, microbial growth), plus a set of factors that may have greater potential impact (soil type, bulk density, previous land use, etc). For the case of soil organic C, this model seems very naive.

=> Thanks for your constructive comment here. Actually, we are interested that whether and how stand characteristic affect SOC stock. We will drop SOC component from our analysis, as explained in the above response. We believe that focusing on AGC stock by testing several SEM models in order to clarify the ecological mechanisms may be sufficient for this study.

Specific comments Line 123. Replace "in accordance to" by "regarding the" or "about the" Line 230. Delete "in" Line 247. Please modify to ".. diversities were calculated for each plot using equation 3". Line 254. Replace "analysis" by "calculation" Line 512. Replace by "effect"

=> We will correct the above mistakes in the revised MS. Thank you.

———————————————————

a

[Figure]

b

[Figure]

c

[Figure]

**Fig. 1.**

---

## Author Response (AR1)

**Response to anonymous referee #1**

*Ali et al. present a study on an interesting and important topic: biomass estimation for subtropical forests in the East Asian monsoon region. The study is generally well introduced and clearly structured. The data set is most probably appropriate to tackle the research questions raised by the authors. The choice of analytical methods, however, needs considerable reconsideration in some regards.*

=> We are grateful to referee #1 for providing useful comments on our study. We have thoroughly revised our manuscript (MS) by following the reviewer's suggestions. According to the reviewer's constructive comments, we have reorganized the conceptual models (see Fig. 1 in the revised MS). In addition, we have re-analyzed our data with structural equation models (SEMs) and we believe that our MS has substantially been improved.

=> Please find our responses to your specific comments below.

*1) Measurements and calculations of carbon stocks*
*- There are no measurements of carbon stocks, just calculations based on allometric equations, so please adjust the section title accordingly.*

=> We have adjusted the section title. Thank you.

*- I was not able to find eqn 1 in Brown et al. 1989, please indicate exact reference or modification if applicable.*

=> Actually, we used the revised form of the equation in Brown et al. (1989), which had been published in FAO papers (1997). We apologize for the wrong citation. In the revised MS, we have calculated AGB using equations in Chave et al. (2014), and used the D and H model.

*- 14% of variance in tree height are not explained by diameter. This information could be used to improve allometric estimates, since the diameter-height-allometry varies with environmental conditions, and might provide valuable additional information.*

=> This is a constructive comment. In the revision, we have employed Chave et al. (2014) model by using DBH, H and wood density as predictors, and we believe that this model improved the estimation of AGB of large trees.

*- However, there is no way of validating your AGB estimates, since no yield data are available. In the same regard, the comparison of eqn 1 with other allometric equations is not useful, since you never know the true AGB for the plots. If this comparison shall be kept, then please change it into some kind of uncertainty estimate. Rˆ2 values do not help much here, since all equations are based on the same parameter (diameter), so please report RMSE values. Related: in fig. S3, please provide equidistant scaling of the axes.*

=> We agree with your comment that we cannot validate AGB estimates in the previous MS. We have used the most recent global allometric equation developed by Chave et al. (2014) for estimation of AGB (as recommended by referee# 2), as it has been found to be the most suitable and appropriate equation for tropical and subtropical forests. Therefore, there will be no need to compare AGB estimates from different allometric equations, as allometric equations in Chave et al. (2014) include subtropical forests. Thank you.

*- L191 ff: To me, it is unclear how to relate the DBH of a single tree to area-based basal area estimate. Please elaborate here.*

=> Sorry for the lack of clarity in the previous version of our manuscript. Tree basal area is calculated as pi*(DBH/2)^2, and stand basal area is the sum of all tree basal area. In the revised MS, we have deleted these sentences, as there is no need for comparison anymore. Thank you.

*- L197: You are not using a D-H model.*

=> We have clarified this in the revised MS, by using the D-H model for both big trees and small trees and shrubs. Thank you.

*2) Calculation of structural diversity*
*- L210ff: Why do you optimise for a good correlation between H for DBH and height?*
*If you so, you might as well use only one of these factors as a surrogate variable for general tree dimension diversity. I suggest comparing results for different discretization cutoffs instead. This would also interesting for the SEM approach: stand age drives structural diversity, but the direct link between stand age and C-stocks is stronger than the indirect one. One reason for this might be a mismatch in classification resolution.*

=> We agree with the suggestion and have compared results for different discretization by employing SEMs and select the best SEM through AIC. Please see Table S3 for such

comparisons and selection of best SEM. Moreover, in the revision, we have used stand structural diversity as a latent variable by incorporating both DBH and height diversity indices.

*3) Statistical analysis*

*- You present a variety of linear modeling variants, when all you want to know is how a set of six parameters influences two response variables. The first set of analysis is contained in the second set, and the second set is a complicated way of doing an AIC based stepwise procedure (under the assumption that collinearity in the design matrix is manageable, which you suggest, but might want to reconsider given the explained variance of the single predictors sum up to > 160% (see L330ff)).*

*- The basic question, as I understand it, is: which set of variables is the best choice for predicting C-stocks. Following this logic, a validation approach would be suited to address the problem, either using a stepwise procedure, using explicit variants of multiple regression models (like already done for the second stream of analyses), or a learning routine that allows for inspection of relative variable importance (like random forests). 80 plots could well be enough for such a validation scheme.*

=> Thanks for the constructive comments here. We have followed the comments on diversities and compare the results. Therefore, we have only use SEMs for comparing different models based on different combinations of DBH and height diversities of different discrete classes. In addition, we have provided bivariate relationships and Pearson's correlation coefficients in Fig. 2 and Table S2, respectively. Further, we have also refined our conceptual model in order to test the complex pathways in one SEM model, instead of in two models (as conducted in the previous MS).

*The results are presented in a clear and concise fashion, and the discussion is consistent, comprehensible and linked to current literature, given the results based on the complex analysis scheme.*

=> Thanks a lot!

*Some minor corrections:*

> *- L339 "range" instead of "ranged"*
> *- L480 "which was also found"*
> *- L537 "to increase C storage"*
> *- L187 "using Brown's"*

*- L190 why switch from DBH to D?*

*- L192 "using Brown's"*

*- L194 "that Brown's"*

115 *- L201 AGBt*

*- L247 "using equation 3"*

=> We have corrected all these mistakes in the revised MS. Thank you.

 **Response to anonymous referee #2**

*General comments*
*In general, I consider the MS has great potential in providing a strong contribution to*
*ecological literature by assessing the relative role of different predictors and particularly of*
 *structural and species diversity on carbon stocks in subtropical secondary forests. This is a*
*topic of active research today. However, I consider the current version is still away from*
*publishable in Biogeosciences. I have five main comments on this:*

=> We are grateful to referee #2 for providing constructive comments on our manuscript.
 According to the reviewer's comments, we have thoroughly revised the MS both in
theoretical and analytical aspects. Please find our responses to your specific comments
below.

*1) First, Rather than providing a strong conceptual approach for framing their aim, that is,*
 *testing the role of structural diversity on aboveground biomass, authors made a long but not*
*structured literature review of the many variables that could explain variation in AGC stocks,*
*of course making particular emphasis on those the will further test. After such review, there*
*are no clear stated hypothesis guiding the application of statistical methods and their*
*prediction is so general and non-exclusive that it could be demonstrable almost with any*
 *result. I consider the conceptual model in Figure 1 is a good starting point, but such a model*
*should be clearly sustained in the introduction. It could serve as the hypothesis to be tested.*
*Another argument on favor of this critique is that soil carbon stocks are almost no introduced*
*and furthermore, authors pretend to explain them with the same set of predictors than used*
*for the AGC case. This shows a naive approach that does not take into account the vast*
 *literature on the factors influencing C stocks in (tropical) soils.*

=> Thanks for these constructive comments. We agreed with your concerns that the
research aims are not well structured in our previous MS. In the revision, we have clearly
introduced our new conceptual models in the introduction for driving the specific hypothesis.
 In the introduction, we have argued that stand structural diversity contributes directly to AGB,
but variations in stand structure may also enhance light capture and C storage. Hence, stand
structural diversity may vary more strongly than species diversity within communities (due to
disturbances) and across communities (due to environmental gradients), and may have a
larger direct effect on aboveground C storage (Poorter et al., 2015). Therefore, we
 hypothesized that stand structural diversity would have a stronger and positive effect on
aboveground C storage than species diversity, once the direct effect of stand age has

explicitly been taken into account, in secondary subtropical forests (see conceptual models in Fig. 1).

160 => After careful consideration, we feel that it may be best to exclude the SOC component since data associated with many drivers such as local site condition, past disturbance history as well as litterfall (leaves and roots) feedback for SOC are not available. We have used much of our efforts on aboveground C storage by testing 48 structural equation models (SEMs), in order to clarify the effects of stand age, stand structural diversity and species
165 diversity on aboveground C storage. Therefore, the SOC component has been excluded from our revised MS. Thank you.

*2) In accordance with the unstructured introduction, authors present a wide range of statistical tests for testing basically the same idea. They use simple linear regression,*
170 *multiple regression and SEM to test the same predictors each time. If you have worked to present a conceptual model like that in Figure 1, why to use approximations do does not allow to test it? Moreover, simple and multiple regressions ended providing almost the same results that SEM, with the exception of two new significant interactions in the SEM model, which are then undervalued by the authors. So I would suggest that according to the idea of*
175 *a very clearly presented unique hypothesis, a unique analysis should be presented, in which case SEM seems to be the best option.*

=> We agreed with your comments about statistical analysis. We have used SEMs by testing different combinations of height and DBH diversities based on different discrete classes, and
180 then select the best model through AIC. In this way, we believe that our proposed hypothesis and conceptual model have substantially been improved than the previous version of the MS. We have provided bivariate relationships for each hypothesized path in SEMs in Fig. 2 and correlation coefficients in Table S2.

185 *3) There are some parts of the discussion where authors present possible explanations to their results, but they do not realize that their own results (particularly the SEM) provide no support for such explanations. I consider that a more careful interpretation of such a model should be done.*

190 => We apologize for the lack of clarity in the discussion section in the previous version of the MS. We have now clearly discussed our new model with sound evidences in this and other studies. Thank you.

*4) Authors sometimes cite references that are not appropriate or even not refer to the point under discussion. See several specific comments below.*

=> We have avoided such mistakes in the revised MS. We apologize for inappropriate citations.

*5) I consider the inclusion of site productivity as a predictor should be reconsidered (see specific comments below).*

=> Thanks for your constructive comment. We have followed your suggested paper (Grace et al., 2016) for making a new conceptual model (see Fig. 1). By considering one of your comments below, we have excluded site productivity as a predictor, in the revised models.

*Line 54. Replace ", and store" by "by capturing ". Yu et al 2014 highlight the capture capability rather than the currents C stocks.*

=> We have corrected it in the revised MS. Thank you.

*Lines 58-59. Authors assert site productivity impact C stocks. However, Lohbeck et al. did't tested the effect of site productivity on biomass or carbon stocks, they tested the reverse. A recent test of the effect of productivity on biomass can be found in Grace et al. 2016 Nature for grasslands, or the general hypothesis for the causal relations between productivity and biomass in tropical forests can be found in Quesada et al. 2012 Biogeosciences or Malhi et al 2012 J. of Ecology*

=> Thanks for pointing it out. Actually, this is a wrong citation. We have corrected the problem in the revised MS. Your suggested papers have been considered while making a new conceptual model. Thank you.

*Line 60. Does species diversity impact C stocks? The reference provided (Con et al. 2013) does not seem to provide conclusive evidence. I suggest to soften this assertion and to look for additional literature to sustain it. See for example Cardinale et al. 2011. Am. J. Bot.*

=> Revised as recommended. Thank you.

230 *Lines 61-63. Although authors use consistently a definition of "stand structural characteristics" throughout the MS which includes both "structural" and "diversity" variables, I consider this concept does not provide to the reader a complete idea of what is being tested here, and could hamper the interest on the work. The role of biodiversity has been the subject of much research in the last two decades and stating it separately may make more*

235 *appealing the work to a broader audience. Therefore, I would suggest to use different concepts for structure and diversity.*

=> This comment is very constructive. We have followed your suggestion by considering stand structural diversity as a latent variable including DBH and height diversity, while

240 species diversity as a separate variable, as shown in the conceptual models (Fig. 1). Thank you.

*Line 66. Include the recent work from Poorter et al. 2016 in Nature "Biomass resilience of secondary forests"*

245

=> We have included their work. Thank you.

*Line 69. I would say that Age is a variable that summarizes or reflects the action of several processes. Probably the authors need to rethink how age is included in their conceptual*

250 *model. Particullarly, which would be the direct effect of stand age on carbon stocks? What is the ecological mechanisms behind such effect?*

=> Indeed, Age is a variable that is related to processes such as growth, ingrowth and mortality. Our data do not include process-based measurements, and we wish to use age to

255 summarize multiple processes responsible for standing aboveground carbon. We have used a complex conceptual model in the revised MS. In the previous version of the MS, using two different models, such as age model and stand characteristics model, caused much confusion. We have avoided such type of confusion in the revised model. Thank you.

260 *Lines 78-82. Soil C is an important component of the study. However, it is just briefly introduced and the ecological mechanisms linking aboveground biomass or productivity with soil C stocks are not explained here. Therefore, your questions regarding soil C are not fully understandable.*

265 => After careful consideration, we feel that it may be best to exclude the SOC component since data associated with many drivers such as local site condition, past disturbance history

as well as litterfall (leaves and roots) feedback for SOC are not available. Therefore, the SOC component has been excluded in the revised MS.

270  *Lines 83-90. These lines say the same than previous paragraphs, no? Probably better to merge them with previous paragraphs and to try to focus more on the general hypothesis regarding the effects of forest age, stand structure and stand diversity.*

=> We have revised and rearranged our introduction by basing on new conceptual models,
275  as you have suggested in earlier comments. We have proposed a new hypothesis based on our new conceptual models. Thank you.

*Line 98. What is C synthesis?*

280  => We apologize for using different terminology here. Actually, we meant C stock or storage.

*Lines 110-111. So anything could explain C stocks? Isn0t there a hypothesis on which of this potential explanatory variables could be more important? Also, what is stand density? Isn't it included within stand structure in general?*

285

=> Thanks for your constructive comment. In the revised MS, we have considered stand structural diversity including DBH and height diversity, and species diversity as potential explanatory variables, when assessing the residual effect of stand age on both of them. Stand density is the number of trees per hectare. Yes, it is included within stand structure in
290  general. We have avoided this variable in the revised conceptual models.

*Line 112. What is a direct effect of stand age? Isn0t it mediated always by stand charachteristics? Which is its ecological basis?*

295  => With increasing stand age, biomass accumulation will increase by following stand development, tree growth and increased stand structural diversity. Therefore, stand age can act as a driver for increasing carbon stocks. In the revised MS, we used one complex conceptual model. In the previous version, using two different models, such as age model and stand characteristics model, caused much confusion. We have avoided such type of
300  confusion in the revised model. Thank you.

*Lines 114-115. This generalization applies only for wet forest, probably not for dry forests. Please be specific.*

305 => We have considered the general approach here (Bazzaz, 1979), by considering the original reference in the revised MS. However, this generalization also applies for dry forests but probably based on different aspects of ecological mechanisms (Becknell and Powers, 2014). Thank you.

310 *Lines 117-118. That is not an adequate prediction, that is a "all matters" scenario. Rather, say that you tested the contribution of different predictors.*

=> We have revised here according to our new conceptual models. Thank you.

315 *Line 122. Randomly? Within the entire landscape? How were you sure they represented all the successional gradient possible? There were no mature forests, conserved and/or degraded? Did you use a GIS to select them? Please elaborate on site selection.*

=> Thanks for pointing it out. 'Randomly' is not an appropriate description of site selection.
320 Actually, we selected sites and plots through both field survey and local forestry inventory that were used for classifying regional vegetation types. We have further elaborated on site selection in the revised MS. Thank you.

*Line 122. Stand age in relation to what? What kind of disturbance?*

325

=> We defined stand age as time since last stand replacing disturbance, which includes clearcutting, reclamation from agriculture, and windthrow by typhoon. This has been clarified.

330 *Lines 124-130. Questions should be rephrased, their actual form is not appealing (they seem barely descriptive). Also, questions 1 and 2 are the same but in their discrete and continuous forms, respectively.*

=> We have revised the proposed questions according to the new hypothesis and
335 conceptual models. Thank you.

*Line 140. The "consequently" is not clear. Authors asserted "there were different intensities of human disturbances (typically logging)" Do they refer to different types of disturbance, different intensities of logging, or both? This is quite important since recent studies on*
340 *succession have highlighted the relevance of different types of previous land-use or land-use*

*intensities for the unfold of succession (Mesquita et al. BioSciences 2015, Arroyo-Rodríguez 2015 Biological Reviews). Moreover, it is particullarlly relevant the authors provide a detailed description of the disturbance history of the region and of the related criteria for selecting plots in particular.*

345

=> Yes, different types of disturbances such as logging, land conversion, windthrow by typhoon etc, as well as different intensities of logging at different sites happened in the history. We have clarified those in the revised MS. Thank you.

350 *Line 141. Rather than developmental stages, which may refer to a departure from a clear-cutted forests, authors could use "stands with different levels of degradation" or "stands with different level of perturbation"*

=> We have revised it as recommended. Thank you.

355

*Line 142. Does this mean that there was previously a landscape characterization of different landcover types from which it was possible to filter only successional forests and to select randomly the location of the plots?*

360 => Yes, more exactly saying, there was a landscape characterization of different forest use types, i.e., secondary shrublands, mature forests protected from clearcutting or logging, and logged forest. We have clarified this section "Study site, plots and forest structure" in the revised MS. Actually, the detailed description of the study area was not included in the previous version.

365

*Line 143. Any kind of disturbance? Excluding only recent human disturbance? What do authors mean exactly by "recent"?*

=> We have clarified the kind of disturbance in the revision. Recent means for the last 3
370 decades according to records from the local government. Thank you.

*Line 148. What do the authors mean by "typical habitats"? Did the authors include plots in different environmental conditions? Or do they refer to different successional habitats all in under the same environmental conditions?*

375

=> Sorry for the vague wording, we have rephrased this statement. Thank you.

*Line 152. It is interesting that until here I assumed the authors constructed a chronosequence of sites derived from a pulse-type disturbance. This was probably because of the use of the terms forest age and secondary forests, which are commonly used in the literature to refer to clear-cutted sites. However, after looking at Table 1, I figured out that sites were assigned to one of three different "development stages", which seems to be different in the intensity of previous logging. Therefore, sites were not clear-cutted but instead affected by a pressure-disturbance like continuous logging. Therefore, I suggest the authors provide their working definition of secondary forest, or, alternatively, use the term "degradation level", "degradation intensity" or simply "logging intensity" to refer to their different levels of logging. Authors can look at several references for the definitions of secondary forest and degraded forest (Chazdon 2014 Second Growth, Chapter 1; Chokkalingam & de Jong 2001 International Forestry Review; Putz & Redford 2010 Biotropica).*

=> Thanks for your constructive and helpful comments on site selection. We have clarified those in the revised MS, by following your comments and suggested papers for definitions.

*Line 169. Which stages? You have not defined such stages here.*

=> Developmental stages such as young, pre-mature and mature forests. Now, we have changed this term to "stands with different levels of degradation", as suggested.

*Lines 170-171. Ok, so it is an indirect measure of productivity. Much more is therefore required on the definition of the disturbance regime to which such plots were subjected. Was the initial point (year 0) a clear-cutted forest for all? Or a selectively logged forest as suggested by Table 1?*

=> Yes, we have indirectly estimated the site productivity by reviewing the official documents of Ningbo Forestry Bureau, Zhejiang Province, to collect relevant data about the disturbances for each site in the study area. The study plots included both clear-cut forests and selectively logged forests. More specifically, there was a landscape characterization of different forest use types, i.e., secondary shrublands, mature forests protected from clearcutting or logging, and logged forest. Site productivity as a predictor has been excluded from new analyses, as recommended by the reviewer.

*Line 176. Which one of these references was used to calculate biomass? Please be specific.*

415 => We used both references because Brown's (1989) equation only covers trees with DBH > 5 cm while equations in Ali et al. (2015) were developed for small trees and shrubs. Thank you.

*Lines 175-184. Why is this paragraph here? A portion could be used during model framing in*
420 *the introduction section.*

=> Thanks for the constructive suggestion here. We have deleted all description about site productivity, as site productivity is not included as a predictor in the revised MS.

425 *Lines 188-189. This is not an argument to exclude height from biomass calculation.*
*See for example Chave et al. 2014 GCB for a detailed discussion on height inclusion in*
*allometric equations.*

=> We have used recent general allometric equations using DBH, H and species' wood
430 density as predictors for the calculation of AGB (Chave et al., 2014) in the revised MS. Thank you.

*Line 192. First sentence is not clear: what kind of uncertainty is avoided and why?*
*Line 193. second sentence should be re-written*

435

=> Actually, most of the generalized allometric equations are for tropical forests instead of subtropical forests. Therefore, we compared different models to avoid uncertainty. We have used the Chave et al. (2014) equation, which includes subtropical regions. Thank you.

440 *Line 196-197. what are D-H models?*

=> Model using DBH and height as predictors for estimation of AGB. We have clarified this in the revised MS.

445 *Lines 210-211. Why you did not use the Chave et al. 2014 equation, which seems to*
*improve Chave0s et al 2005 equations?*

=> We have used the Chave et al. (2014) equation in the revised MS. Thank you.

*Line 2015. Therefore, which equation you used? I suggest all this discussion could go in an Annex or supplementary material, leaving here in the methods only the description of the equation finnally used*

=> Brown's equation was used for the estimation of AGB of big trees. Now, we have used the Chave et al. (2014) equation in the revised MS. Thank you.

*Line 216. Why you did not used the Alí et al. 2015 equation for all the tree community?*

=> Ali et al. (2015) equations were only developed for small trees and shrubs.

*Line 239. Does this values refers to the number of categories, the range of the categories or the limits of the categories?*

=> These values refers to the limits of the categories. For example, for DBH < 2 cm, 2.1 – 4 cm, 4.1 – 6 cm, etc.

*Lines 244-245. Why to use correlated DBH-height classes if you then want to assess their explanatory ability in a unique multiple regression model? Should not the categories be selected based on their correlation to the variables you want to explain, i.e. biomass? You could simply try to test correlation between diversity and biomass and select those categorizations given the maximum correlation.*

=> Thanks for your constructive comment. By following this comment, we cannot get any good fit for the SEM model when we tried. Therefore, it is better to test different SEM models instead of just focusing on correlations. In the revision, we have tested a number of SEM models through combinations of different DBH and height diversities based on different discrete classes, and then select the best model based AIC (see Table S3). In order to make things clearer, we have provide statistics of all SEM models in Table S3 and more details for selected best models (Fig. 3; Table 1).

=> Interestingly, when we used correlation between DBH and height diversity as a latent variable 'stand structural diversity' in SEMs, we also cannot get a good model fit, indicating that these two diversities are independent in our study. Thank you.

*Line 251. Mathematical notation is wrong. x should denote only one thing: or the number of different attributes evaluated (3) or the number of classes within a attribute.*

*Furthermore, sub-index for p should be i (pi), because the proportion is evaluated for each i class within 1 and x (if x is the total number of classes).*

=> We have corrected the equation form in the revised MS. Thank you for pointing it out.

*Line 270. Please say explicitely at the beggining of the secition 2.3 which C pools are considered in this study: "two carbon pools were assessed in this study: aboveground living biomass of the tree community (excluding lianas and herbs, no'), and soil organic C in the top 20 cm of soil").*

=> We have clarified it in the revision.

*Lines 270-276. Probably better to summarize lines 270-276 by saying that for each series, al the possible variable combinations and interactions were tested (a fully ...model) and the best model was selected using AIC.*

    *Line 291. If you have previously settled a hypothesis of a hierarchy of effects acting onC stocks, why to use simple and multiple linear models and not going directly to the SEM? What is the original hypothesis? Doesn0t SEM allows you to test the same that multiple regression model allows, that is, which are the structural determinants of the C stocks?*

=> By considering all of your comments on the conceptual model, we have only employed SEMs in the revised MS. In addition, the bivariate relationships are included in Fig. 2 and correlation coefficients in Table S2. Thank you.

*Line 304. Age is not expected to be linearly related to AGC. Also, from Figure 2 it seems that some of the relations could be better explained using a non-linear (but probabliy linearizable) model.*

=> We have considered your suggestion in the revised MS, by assessing both linear and several linearizable forms (log, $2^{nd}$ and $3^{rd}$ order polynomial). Finally, we used the simple linear regression analysis to test for bivariate relationships because, 1) there were no big differences between linear and non-linear relationships that may cause any big difference in our results; and 2) in order to avoid complexity of the composite variables in the SEMs. Thank you.

*Lines 307-310. So, really the logic behind fitting such models was to select the best to use in SEM? Why not allowing SEM to test the whole model? Why testing two different models if you can test only one?*

=> We have used one SEM model and accessed the whole model as well as the best model based on AIC. Thanks for the helpful suggestion.

*Lines 314-320. This paragraph is very difficult to grasp. Does the second sentence mean that rather the structural diversity, the proportion of big trees could alternatively explain biomass?*

=> Yes, you are right. We have deleted this method in the revised MS because of not too much helpful.

*Lines 315-318. If I understood well, this is the same problem with analyzing Shannon index results for species diversity: we do not actually know if an increased diversity is caused by increased number of categories (which in this case means increased number of big trees) or by a more even distribution among categories (that is, basal area is more equitatively distributed among dbh categories). If you want to dissect such effects, then wouldn0t be easier to have from the beginning to different predictors indicating directly such different possible explanations? Moreover, previous findings would allow authors to hypothesize that the amount of big trees is an important predictor of forest biomass (Slik et al. 2013 Global Ecol. Biogeo.), so authors could use some indicator of the size of the biggest trees as a predictor of biomass.*

=> Thanks for your constructive comments here. We have used SEMs to test different combination of DBH and height diversities based on different discrete classes, to know whether increased diversity caused by increased number of categories has any different effect on aboveground C storage.

*Line 322. I0m not completely sure that a higher correlation with CV means that dominance of big trees is not important. Higher CV values means that deviation from the mean DBH or H increases, which can happen if bigger trees are present but there is an uneven size distribution.*

=> We have used the alternative approach, as you have suggested above. Thank you.

*Line 332. Most of the significant relations seems to violate linear regression assumptions, particularlly that the straight line is an adequate representation of the relationchip or that variance is homogeneous. Authors do not clarify through the text or in the supplementary tables if other relationships were tested or if variables were transformed to meet assumptions*

=> We have provided details in the revised MS, please see the third paragraph in the statistical analyses in the revised MS. Thank you.

*Line 334. Species density? Stand density?*

=> Actually, it was species diversity and stand density (trees per hectare). We have clarified this.

*Line 341. What is the positive variation?*

=> Means positive linear relationship. We have revised this.

*Lines 360-363. Probably, the synthetic models are not necesary. Authors can check that the relative importance of variables in the synthetic model correlates negatively but perfectly to the p values associated to each of the variables in the best-fit model. So probably that part could be taken to the supplementary material.*

=> Thanks for the constructive comment here.

*Lines 368-369. As expected, there is no direct functional relation between stand characteristics and C stocks. This only reflects the poor literature review on the mechanisms that drive C accumulation in tropical forests soils.*

=> We have included more potential and recent literature about aboveground C storage, while we have dropped the SOC component in the revised MS. Thank you.

*Lines 377-379. There is no sense in having these two alternative models, at least if there are no competing hypothesis grounded on strong ecological knowledge.*

=> We have focused on our new conceptual model. Therefore, this part has been updated.

*Lines 380-381. I really have a doubt on the meaning of the variable "productivity" here. As defined, productivity is calculated on the basis of stand volume divided by forest age. Stand volume is another measure of biomass (the volume of a forest is filled with biomass, so as it is bigger, biomass is bigger), rather than an "independent" structural measurement. I really think that it is an spurious relation and that the authors should consider to exclude it from the model.*

=> We agreed with your suggestion to exclude productivity from our conceptual model. Thank you.

*Lines 382-383. What is the difference between this model and the multiple regression model?*

=> Sorry for providing double proof of the results. We have only considered SEMs in the revised MS.

*Lines 410-411. This last sentence evidence the poor literature review made by the authors on the ecological and physical processes controlling C stocks in soils. I suggest to not include soil C stock estimation in the model, but rather to provide their estimates as supplementary material.*

=> We have excluded SOC in the revised MS. Thank you.

*Lines 419-420. Such argument would imply that higher species diversity have incidence on higher structural diversity. However, there is no association between species and DBH diversity, so data does not support such possibility.*

=> According our new analysis in SEM, it is clarified now. Please see Table 1 and Fig. 3 for positive association between species and stand structural diversity. Thank you.

*Line 433. If such argument was true, a significant relation between species diversity and stand age should arise.*

=> We have revised it accordingly. Please see lines 1441-1454 in the revised MS.

*Line 449. Uncertain? It seems authors are "averaging" results from two different approaches and therefore saying that there is no conclusive evidence, even with the same data! That0s*

*why it is important to have a clearly stated hypothesis from the beginning and to use the*

635 *adequate analytical framework to test it.*

=> This section has been updated by focusing on our new conceptual models. Thank you.

*Line 451. A similar argument was raised by Grace et al. 2016 Nature*

640

=> Thanks for your interest in the argument but this section has been updated based on our new analysis.

*Lines 467-468. Site productivity does not mediate such relation according to SEM.*

645 *Please rephrase.*

=> This section has been updated based on our new analysis. Thank you.

*Line 481. Dupuy et al. 2012 do not test age as a predictor of biomass. Please see*

650 *Hernández-Stefanoni et al. 2010 Landscape Ecology for the adequate reference.*
*There are a lot more of references on the recovery of biomass or AGC stock during*
*succession in both wet and dry tropical forests. See also Poorter et al. 2016 Nature for*
*a recent compendium.*

655 => We have updated it by citing most recent studies (Poorter et al., 2016). Thank you.

*Lines 485-487. this argument is not right. Although it is true that at tree level bigger trees*
*acumulate more carbon, at the stand level it is not true if we have a gradient of forest age,*
*for which maximum accumulation commonly occurs early in succession.*
660 *See Mora et al. 2016 Biotropica, Vargas et al 2008 GCB or Yang et al. 2011 New*
*Phytologist for how expected rates of change should be higher in the first decades of*
*succession.*

=> We have deleted this sentence in the revised MS because we are focusing on the stand
665 level analysis instead of tree level. However, we have provided argument to support our
result in lines 746-748.

*Line 488. Not pretty sure of this since CV test does not seem to be the best indicator.*

670 => For CV of DBH as a good predictor of AGB, please see (Zhang and Chen, 2015).

However, we have not focused on CV in the revised MS. Thank you.

*Line 499. Lohbeck et al. 2015 never tested productivity as a predictor of biomass, but the reverse (biomass as a predictor of productivity).*

=> We have revised it. Thank you.

*Lines 500-504. In the model site productivity is not affected by forest age, so this argument does not march data.*

=> We have excluded productivity from our conceptual model, as you have suggested in an earlier comment. Thank you.

*Line 514-516. This argument is not clear at all*

=> We have clarified it in the revised MS.

*Lines 536-537. Please elaborate more on how stand diversity could be improved based on your results.*

=> We have elaborated it in the revised MS. Thank you.

*Line 790. Why should soil organic C depend on structural stand variables? There are many ecological process between C accumulation in the aboveground biomass and its accumulation in soil (literfall, biomass decay, microbial growth), plus a set of factors that may have greater potential impact (soil type, bulk density, previous land use, etc).*
*For the case of soil organic C, this model seems very naive.*

=> Thanks for your constructive comment here. Actually, we were interested that whether and how stand characteristic affect SOC stock. We have dropped the SOC component from our analysis, as explained in earlier responses.

*Specific comments*
*Line 123. Replace "in accordance to" by "regarding the" or "about the"*
*Line 230. Delete "in"*
*Line 247. Please modify to ".. diversities were calculated for each plot using equation 3".*

*Line 254. Replace "analysis" by "calculation"*

*Line 512. Replace by "effect"*

710

=> We have corrected the above mistakes in the revised MS. Thank you.

*References*

715 Bazzaz, F.: The physiological ecology of plant succession, Annual review of Ecology and
Systematics, 10, 351-371, 1979.

[revised manuscript text omitted]

49 importance (Imp.), regression coefficient (Coeff.) and standardized regression coefficient (Beta) are given. The results are averaged over all

50 seven possible models using AICc-wi (the Akaike information criterion weight) as a weighting criterion for first and second series, but averaged

51 over all 63 possible models using AICc-wi for third series. Significant coefficients ($P < 0.05$) are given in bold.

| Predictor variable | Synthetic model of first series | | | Synthetic model of second series | | | Synthetic model of third series | | |
|---|---|---|---|---|---|---|---|---|---|
| | Imp. | Coeff. | Beta | Imp. | Coeff. | Beta | Imp. | Coeff. | Beta |
| Constant | --- | -14.26 | 0.00 | --- | -29.71 | 0.00 | --- | -5.70 | 0.00 |
| Species diversity | 0.97 | -23.54 | **-0.24** | | | | 0.96 | -15.98 | **-0.16** |
| Height diversity | 0.28 | -8.50 | -0.06 | | | | 0.77 | -21.03 | **-0.14** |
| DBH diversity | 1.00 | 106.23 | **0.75** | | | | 0.90 | 30.13 | **0.21** |
| Stand age | | | | 1.00 | 0.81 | **0.61** | 1 | 0.78 | **0.59** |
| Site productivity | | | | 1.00 | 14.50 | **0.56** | 1 | 12.05 | **0.47** |
| Stand density | | | | 0.70 | -0.002 | **-0.11** | 0.42 | -0.002 | -0.08 |

61  **Table S4.** The best model obtained from a series of regression analyses of a response variable (soil organic C stock) on each of stand structural

62  diversity (species, DBH and height diversity; first series), other stand characteristics (stand sage, stand density and site productivity; second

63  series), and a combination of stand structural diversity and other stand characteristics (third series). For each predictor variable, the regression

64  coefficient (Coeff.), standardized regression coefficient (Beta), $t$-test and $P$-value are given. The coefficient of determination ($R^2$), $F$-test, $P$-

65  value and Akaike Information Criterion (AICc) of the model are also given. For each effect of first and second series, all 7 possible models were

66  tested, while all 63 possible models were tested for third series. See Table S5 for the contribution to the models of all variables tested. Detailed

67  statistics of all models for first, second and third series are provided in Tables S12, S13 and S14, respectively. $P$ values < 0.05 are given in bold.

| Model and predictor variable | Coeff. | Beta | $t$ | $P$ | $R^2$ | AICc |
|---|---|---|---|---|---|---|
| **Effects of stand structural diversity** | | | | | | |
| *Model*[1] | | | 0.10 | 0.751 | 0.002 | 611.75 |
| Constant | 80.46 | 0.00 | 6.25 | **<0.001** | | |
| Height diversity | 2.69 | 0.04 | 0.32 | 0.751 | | |
| **Effects of other stand characteristics** | | | | | | |
| *Model* | | | 0.99 | 0.324 | 0.02 | 610.84 |
| Constant | 90.70 | 0.00 | 12.83 | **<0.001** | | |
| Site productivity | -1.88 | -0.12 | -0.99 | 0.324 | | |
| **Joint effect of stand structural diversity and other characteristics** | | | | | | |
| *Model* | | | 0.99 | 0.324 | 0.02 | 610.84 |

| | | | | | |
|---|---|---|---|---|---|
| Constant | 90.70 | 0.00 | 12.83 | <0.001 | 68 |
| Site productivity | -1.88 | -0.12 | -0.99 | 0.324 | 69 |

[1] The value under $t$ column represents $F$-test of the model

71 **Table S5.** Synthetic model obtained from a series of regression analyses of a response variable (soil organic C stock) on each of stand structural

72 diversity (species, DBH and height diversity; first series), other stand characteristics (stand sage, stand density and site productivity; second

73 series), and a combination of stand structural diversity and other stand characteristics (third series). For each predictor variable, their importance

74 (Imp.), regression coefficient (Coeff.) and standardized regression coefficient (Beta) are given. The results are averaged over all seven possible

75 models using AICc-wi (the Akaike information criterion weight) as a weighting criterion for first and second series, but averaged over all 63

76 possible models using AICc-wi for third series. Significant coefficients ($P < 0.05$) are given in bold.

| Predictor variable | Synthetic model of first series | | | Synthetic model of second series | | | Synthetic model of third series | | |
|---|---|---|---|---|---|---|---|---|---|
| | Imp. | Coeff. | Beta | Imp. | Coeff. | Beta | Imp. | Coeff. | Beta |
| Constant | — | 83.10 | 0.00 | — | 88.70 | 0.00 | — | 88.34 | 0.00 |
| Species diversity | 0.43 | -0.69 | -0.01 | | | | 0.29 | -1.58 | -0.03 |
| Height diversity | 0.44 | 2.92 | 0.04 | | | | 0.29 | 1.59 | 0.02 |
| DBH diversity | 0.43 | -0.13 | -0.002 | | | | 0.29 | -0.93 | -0.01 |
| Stand age | | | | 0.39 | 0.04 | 0.06 | 0.30 | 0.04 | 0.06 |
| Site productivity | | | | 0.50 | -1.80 | -0.12 | 0.40 | -1.93 | -0.13 |
| Stand density | | | | 0.47 | -0.001 | -0.11 | 0.36 | -0.001 | -0.11 |

79 **Table S6.** Collinearity statistics for each characteristics of the stand within multiple regressions model of each of aboveground C and soil

80 organic C stock.

| Predictor variables | Collinearity Statistics | |
| --- | --- | --- |
| | Tolerance | VIF |
| *Aboveground C stock as a response variable* | | |
| Species diversity | 0.83 | 1.20 |
| Height diversity | 0.56 | 1.79 |
| DBH diversity | 0.36 | 2.78 |
| Site productivity | 0.58 | 1.71 |
| Stand age | 0.51 | 1.95 |
| Stand density | 0.68 | 1.47 |
| *Soil C stock as a response variable* | | |
| Species diversity | 0.79 | 1.27 |
| Height diversity | 0.52 | 1.94 |
| DBH diversity | 0.39 | 2.56 |
| Site productivity | 0.65 | 1.54 |
| Stand age | 0.51 | 1.96 |

| | | |
|---|---|---|
| Stand density | 0.65 | 1.55 |

82 **Table S7.** Direct, indirect, and total standardized effects of predictors on aboveground C stock based on structural equation models (SEMs). The

83 upper section of table (model A) showing the direct standardized effect of stand characteristics on aboveground C stock (see Fig. 4a). The lower

84 section of table (model B) showing the direct, indirect, and total effects of stand age on aboveground C stock; and also direct effect on other

85 stand characteristics (see Fig. 4b). Significant effects are at *, P < 0.05; **, P < 0.01; ***, P < 0.001; and ns, not significant.

| SEM model and response variable | Predictors within each model | | | | | | | | | | | | | | | | | |
|---|---|---|---|---|---|---|---|---|---|---|---|---|---|---|---|---|---|---|
| | Stand age | | | Species diversity | | | DBH diversity | | | Height diversity | | | Site productivity | | | Stand density | | |
| | Direct | Indirect | Total | Direct | Indirect | Total | Direct | Indirect | Total | Direct | Indirect | Total | Direct | Indirect | Total | Direct | Indirect | Total |
| **A, stand characteristics model in Fig. 4a** | | | | | | | | | | | | | | | | | | |
| ACS | 0.61 | — | 0.61 | -0.19 | — | -0.19 | 0.24 | — | 0.24 | -0.14 | — | -0.14 | 0.46 | — | 0.46 | — | — | — |
| | *** | | *** | *** | | *** | *** | | *** | * | | * | *** | | *** | | | |
| **B, stand age model in Fig. 4b** | | | | | | | | | | | | | | | | | | |
| ACS | 0.59 | 0.11 | 0.70 | -0.16 | — | -0.16 | 0.21 | — | 0.21 | -0.15 | — | -0.15 | 0.48 | — | 0.48 | -0.08 | — | -0.08 |
| | *** | ns | *** | * | | * | *** | | *** | * | | * | *** | | *** | ns | | ns |
| Species diversity | 0.16 | — | 0.16 | — | — | — | — | — | — | — | — | — | — | — | — | — | — | — |
| | ns | | ns | | | | | | | | | | | | | | | |
| DBH diversity | 0.63 | — | 0.63 | — | — | — | — | — | — | — | — | — | — | — | — | — | — | — |
| | *** | | *** | | | | | | | | | | | | | | | |
| Height diversity | 0.55 | — | 0.55 | — | — | — | — | — | — | — | — | — | — | — | — | — | — | — |
| | *** | | *** | | | | | | | | | | | | | | | |
| Site productivity | 0.15 | — | 0.15 | — | — | — | — | — | — | — | — | — | — | — | — | — | — | — |

| | ns | | ns | | | | | | | | | | | | | | | |
|---|---|---|---|---|---|---|---|---|---|---|---|---|---|---|---|---|---|---|
| Stand density | -0.09 | —— | -0.09 | —— | —— | —— | —— | —— | —— | —— | —— | —— | —— | —— | —— | —— | —— | —— |
| | ns | | ns | | | | | | | | | | | | | | | |

87 **Table S8** Correlation coefficients between stand structural (tree DBH and height) diversity and each of 90-percentile diameter/height and

88 coefficient of variation in diameter/height of trees.

| Stand variables | CV of D | P90 of D | CV of H | P90 of H | |
|---|---|---|---|---|---|
| Tree DBH diversity | 0.681** | 0.243* | 0.322** | 0.073ns | 90 |
| Tree height diversity | 0.433** | 0.127ns | 0.576** | 0.114ns | 91 |

92 CV: coefficient of variance; D: tree diameter; H: tree height; and P90: 90 percentile; indicated with asterisks if statistically significant (*: $P <$

93 0.05; **: $P < 0.01$; ***: $P < 0.001$; ns: not significant)

95  **Figure Legends**

96  **Fig. S1** a) Number of trees per stand (trees $0.04ha^{-1}$); b) stand volume ($m^3 ha^{-1}$); c) stand aboveground biomass ($Mg ha^{-1}$); and d) standard

97  deviation of stand aboveground biomass of 80 subtropical forest plots.

[Figure]

99    **Fig. S2** Relationship between the log of basal area (m² ha⁻¹) and log of diameter at breast height (DBH, cm) for all tree species (DBH > 5cm)

100    across 80 subtropical forest plots.

[Figure]

102 **Fig. S3.** Comparison of the individual tree aboveground biomass (DBH ≥ 5 cm) estimated with Brown's (1989) equation and a) simple

103 geometric equation, and b, c) Chave et al.'s (2005) moist forest equations. The dashed line represents a 1:1 theoretical relationship; the solid line

104 represents the observed relationship. It should be noted here that the individuals having specific wood density were used for the comparison

105 purpose only.

106 For panel (a) of the graph; a simple geometrical equation suggests that the total aboveground biomass (AGB, in kg) of a tree with diameter D

107 should be proportional to the product of wood density ($\rho$, oven-dry wood over green volume), times trunk basal area (BA= $\pi D^2/4$), times total

108 tree height (H). Hence, the following relationship should hold across forests:

109 $AGB = F \times \rho \times (\frac{\pi \times D^2}{4}) \times H$       (a)

110 Dawkins (1961) and Gray (1966) predicted a constant form factor (F) across broadleaf species, with F = 0.06 (Cannell, 1984).

111 For panels (b. c) of the graph, the best predictive models proposed by Chave et al. (2005) for moist forests were used to estimate the AGB (kg)

112 of each individual tree.

113 $AGB = \exp(-2.977 + \ln(\rho D^2 H)$       (b)

114 $AGB = \rho \times \exp(-1.499 + 2.148 \times \ln(D) + 0.207 \times (\ln(D))^2 - 0.0281 \times (\ln(D))^3)$   (c)

115

[Figure]

118    **Fig. S4**

119

120

[Figure]

[Figure]

121

122  **Fig. S5** Best-fit structural equation models for soil organic C stock; a) combining species diversity, DBH diversity, height diversity, stand age,

123  stand density and site productivity (stand characteristics model), b) stand age as a primary predictor variable by testing the direct and indirect

124  effects through mediation of stand characteristics (stand age model), across all 65 subtropical forest plots. Values give the standardized

125  coefficients for the relationship and correlation between variables; all coefficients are significant at *, $P < 0.05$; **, $P < 0.01$; ***, $P < 0.001$; ns,

126  non-significant; and coefficient of determination ($R^2$) for response variable are indicated. Epsilons ($\varepsilon$) within small circle represent the error term

127  for downstream variables, ellipse represents response variable (soil organic C stock), and squares or rectangles represent predictor variables. But

128  in case of model (b), the squares or rectangles with white fill color represent mediators while with gray fill represent primary variable. Model fit

129  statistics for each of the stand characteristics and  stand age models are Chi-square = 4.62 and 7.50, $df$ = 6 and 5, $P$ value = 0.593 and 0.186, CFI

130  = 1.00 and 0.97, GFI = 0.98 and 0.97, CMIN/$df$ = 0.77 and 1.50, RMSEA < 0.001 and = 0.09.

[Figure]

a

b

131

132

---

## Referee Report (RR1)

[Figure]

b

[Figure]

c

[Figure]

**Fig. 2**

[Figure]

**Fig. 3**

[Figure]

[Figure]

[Figure]

35

[referee-annotated manuscript omitted]

---

## Author Response (AR2)

Functional Ecology Lab, 444, R&ES Building
School of Ecological and Environmental Sciences
Tel & Fax: (0086) 21-54341164
E-mail: eryan@des.ecnu.edu.cn

13th July, 2016

Dr. Anja RammigAssociate Editor
Biogeosciences
European Geosciences Union

**Re: Submission of revised manuscript**

Dear Dr. Anja Rammig,

Please find attached our revised manuscript (bg-2016-6) "**Stand structural diversity rather than species diversity enhances aboveground carbon storage in secondary subtropical forests in Eastern China**", submitted as a research paper for *Biogeosciences.*

We are indeed grateful for your patience and helpful suggestions during the review process. We also greatly appreciate the comments from the two reviewers. Please find our point-by-point responses attached in the file "Response to referees".

The coauthors have did revision without track changes in the manuscript. We have highlighted the changes in red color. Meanwhile, we also asked a native speaker (Frank Boelm, c/o NanoApps Consulting, 270 Ray Court, Thunder Bay, ON, Canada) to revise (please find his track changes) the manuscript. As a consequence, our manuscript has been considerably improved.

Yours sincerely,

Dr. En-Rong Yan
Professor in Vegetation and Functional Ecology

**Response to referee #1**

*The manuscript has improved significantly, and all of the issues I raised previously have been addressed. The manuscript is now much more concise and focused. Only one question remains open (see below) and a few language-related issues should be fixed prior to publication.*

=> Thank you for your positive comment on our revised manuscript (MS). We have revised the manuscript according to your comments.

*Model selection: can you explain your model selection strategy in more detail, especially why you prefer models with a Chi-square test resulting in p > 0.05, which seems counterintuitive to me.*

=> We have explained our model selection strategy in more detail in the revised MS. Please see lines 218-221.

*Fig 1.: The difference between the three plots is not directly visible. Have you experimented with different arrow sizes or colors?*

=> We have explained the differences among three models in the caption. Species diversity and stand structural diversity were experimented with based on their different effects on each other. Thank you.

*Language issues:*
*L55: cycling, *and* capture*
*L57: C storages va*ies*, no comma after stand age*
*L68: "teased apart" is maybe not the correct wording; how about "few studies have separated the direct and indirect..."*

*L129: activities for more than the last 25 years*

*L277: Sentence starting with "The maintenance" seems incomplete*

*L310: Our findings of \*a\* weak direct effect*

*L324: and therefore strongly affects aboveground C storage indirectly, via stand structural diversity*

=> We have corrected these mistakes. Thank you.

**Response to referee #2**

*I consider the authors have done a good job dealing with previous comments. The new version of the manuscript is much more clear and focused on what the authors actually can do: test forest attributes as predictors of forest biomass. I consider the manuscript will be a very valuable contribution to the scientific literature on the relation between biodiversity and ecosystem function. However, I still have some concerns about the framing of the problem, the methods, the discussion and the presentation, which I consider deserve further (but minor) work:*

=> We are grateful to you for reviewing our MS once again, and providing constructive comments. We have revised our MS according to your comments, which we believe has substantially improved our MS. Thank you.

*1. Problem statement is still a bit confusing. I provide further suggestions on the attached file to reorganize some ideas/sentences.*

=> We have followed your comments and revised the MS accordingly. Please find below our responses to each specific comment.

*2. Authors still do not provide a clear definition on how was stand age defined. They talk about several kinds of disturbance in the region, but they do not define explicitly in relation to which disturbance was age defined.*

=> We have now explicitly defined stand age. Please see lines 129-130 and 137-140 in the revised MS.

*3. Discussion is yet not fully satisfactory. Particularly, the relevant result on the negative relation between biomass and species diversity is poorly explained. This kind of results have*

*relevant implications for the scientific literature. I provide further suggestions on the attached file.*

=> We have revised the discussion as suggested. Please see lines 264-267 and 282-298 in the revised MS. Thank you.

*4. I think the manuscript requires careful revision by a native English editor to identify and correct some minor mistakes. I have provided some observations through the text.*

=> We have asked a native English speaker to edit the language. Thank you.

*Specific comments:*
*L 48-50. This is not discussed, but merely said in the conclusions section. I suggest to emphasize here the negative effect of diversity.*

=> Revised as suggested. Please see lines 44-55 in the revised MS. Thank you.

*Introduction: Authors have done a well job in focusing the introduction towards the main theme of the MS: the contribution of stand age, species diversity and stand diversity. However, the presentation of the concepts is still confusing. Probably, reordering some of the sentences would improve this section. I suggest they follow this order:*
*1. South East Asian forest are important for C cycling (c capture), probably associated to the high representation of young stands in the region (Yu et al.) C accumulates as forest age, but we still lack a complete understanding of the determinants (mechanisms) of carbon accumulation. Tree species diversity and stand structural diversity have been proposed as factors driving C storage....*
*2. A paragraph on the evidence of stand structural diversity and C stocks and on the evidence of tree diversity and C stocks.*

[Figure]

*3. The interrelation between forest age, structure and tree diversity. Therefore the need to have integrative tests of those mechanisms: Then present their model (Figure 1) and explain it briefly based.*

*4. The aim of the work on how to test the model.*

=> Thank you for your comments. We have followed your suggested theme and revised the entire introduction accordingly.

*L 55. Capturing?*

=> Corrected.

*L 55. Here there is a sentence that is lacking: why the effects of should stand age and species diversity be tested for this forests? Are East Asian forests impacted by human activities? Are secondary forests a common feature of this region?*

=> Revised as suggested. Thank you.

*L 57. ..., and of changes in... Wording is not adequate*

=> Corrected in the revised MS.

*L 58. It should not be cited here*

=> Deleted in the revised MS.

*L 60. The do refer to stand structural diversity, not tree species diversity*

=> Revised

*L 71-80. The paragraph have the main ideas on which the work is based on, but they are poorly presented*

=> Revised as suggested.

*L 78-80. Too simple: why do you expect that? Which direction should it be?*

=> We have now elaborated further in the revised MS. The introduction has been reorganized and rewritten, as suggested.

*L 90-92. You just said the same three lines before*

=> Deleted

*L 93-94. Isn´t it the same in line 74-75?*

=> Deleted.

*L 95. This sentence is the same you have said previously. Please consider moving it to the end of the first paragraph*

=> Revised

*L 103. This is not an adequate citation for SEM, since Grace et al´s paper in Nature is not a methodological paper nor provides new analytical approaches*

=> Revised

*L 105. Hypothesis, represented by paths?*

=> Revised as recommended

*L 129. Odd wording*

=> Corrected.

*L 136-137. Ages are of forest recovery from what kind of disturbance? In the previous paragraph they talked about logging, land-use conversion and wind throw. Is this work mixing plots coming from different kinds of disturbances? If it is, please at least say it explicitly. Moreover, in the response letter...*

=> We have now explicitly stated this in the revised MS. Please see lines 129-130 and 137-140 in the revised MS.

*L 143-144. If plots were at least 100m from stand edges, therefore stands were not as small as authors suggest. I suggest to simply not include any kind of argument relative to plot size.*

=> Revised as suggested. Thank you.

*L 171-175. This should be in the previous section on general site characteristics. And again, it is not clear which disturbance was it.*

=> Revised as suggested. Thank you.

*L 182-183. Sobra*

=> unnecessary phrase deleted

*L 185-186. Why basal area proportions? Why not simply number of individuals?*

=> We have explained this in the revised MS. Please see line 176 in the revised MS.

*L 208-212. But this should be partial correlations to tease apart the independent effects of each variable*

=> Because variables are highly dependent on each other, it was not possible to find independent effects of each variable. Alternatively, we used structural equation models to examine cause-effect relationships among the variables as we hypothesized. We used bivariate relationships to augment SEMs.

*L 244. Strange*

=> revised.

*L 249. Delete*

=> Deleted.

*Discussion: I found the discussion of the two main results disappointing. How can stand structural diversity favor grater C stocks? Authors talk about one mechanisms: enhanced light use. But how does that work? Does higher light use efficiency means greater capacity to stock C? The other relevant results is that on the negative effect of species diversity. Given the central role it has been given to diversity for ecosystem functioning, I think authors should do a better work to argue why the found the reverse. Now the just have a spuriously presented argument of strong functional redundancy. Previous meta-analysis on the effect of species richness on forest AGB (both temperate and tropical) show species richness may have no or negative effect (Cardinale et al. 2011 Amer. J. Bot). Authors should review such previous work.*

=> Thank you for your constructive comments on the discussion section. As suggested, we have thoroughly revised and reorganized the discussion section. By following your suggestions, we believe the MS has been significantly improved.

*L 277-278. Not well written. What do you mean?*

=> Revised.

*L 281. The accumulation of C storage? Isn´t it just C storage? You are assessing C stocks and therefore storage, rather than accumulation (a rate, which you did not measured here).*

=> Revised.

*L 283. In a statistical sense? Your results show it is significant. Moreover, it is negative, contrary to expectations from previous studies!*

=> We apologize for the error. Yes, the effect is significant. We have further discussed the negative effect of species diversity on aboveground C storage.

*L 287-289. Then why other species richer systems shows positive effects of species diversity on AGB (e.g. Poorter et al. 2015)? I find this argument not well sustained*

=> We agree with you and have revised the corresponding text.

*L 292-293. Weak? Compared to what?*

=> Revised and clarified. Please see lines 373-381. Thank you.

L 307. Stand age is not a measure of disturbance frequency

=> Based on the original definition (Connell, 1978), stand age represents disturbance frequency

*L 310-314. Previously, you have said that the negative direct effect of species diversity on AGB may be due to strong functional redundancy. From that, one would expect that removing certain species may favor C accumulation by reducing competition. But here you suggest that selective harvest may have caused the negative association between diversity and AGB. Aren´t those two arguments contradictory?*

=> We agree with you. Species redundancy is not the reason for the lack of diversity effect on aboveground C. We believe that the selective removal of productive species (thus reducing competitive exclusion==high diversity) leads to minimal or negative effects of species diversity on aboveground C. We have carefully revised the Discussion for consistency.

L 319. Related?

=> Corrected.

L 321. Variation

=> Corrected.

L 382. Capitalize first letter

=> Corrected.

Fig. 2 caption. Drop "other"

=> Corrected.

Fig. 2 I suggest to indicate significant relationships with a continuous fitted line, and non-significant with a discontinuous one. Alternatively, do not present the fitted line for those non-significant cases.

=> Revised as suggested. Please see revised Fig. 2.

Fig. 3 caption. How to show that. Please present Chi-2 test results associated to each model

=> Revised as suggested. Please Fig. 3 caption in the revised MS. Thank you.

[revised manuscript text omitted]

b

[Figure]

c

[Figure]

Fig. 2

[Figure]

Fig. 3

a

[Figure]

b

[Figure]

c

[Figure]